# 3DCS: Datasets and Benchmark for Evaluating Conformational Sensitivity in Molecular Representations

**Xi Wang**[1], **Yang Zhang**[2], **Yingjia Zhang**[1], **Yejia Cai**[1], **Shengjie Wang**[1]

{xw3763,yz10442,yc6970,sw5973}@nyu.edu, zhang360428@stu.pku.edu.cn
[1] New York University
[2] Peking University

## Abstract

Molecular representations (MRs) that capture 3D conformations are critical for applications such as reaction prediction, drug design, and material discovery. Yet despite the rapid development of molecular representation models, there is no comprehensive benchmark to evaluate their treatment of 3D conformational information. We introduce 3DCS, the first benchmark for 3D Conformational Sensitivity in MRs. 3DCS evaluates whether representations within the same molecule (i) preserve geometric variation, (ii) capture chirality, and (iii) reflect the energy landscape. To enable this, we curate three large-scale datasets (>1M molecules, ∼10M conformers) spanning relaxed torsional scans, chiral drug candidates, and AIMD trajectories, and propose a unified Geometry–Chirality–Energy (GCE) evaluation framework. Empirical analysis reveals that while modern data-driven MRs are highly geometry-sensitive, they inconsistently handle chirality and poorly align with energy, which is often overlooked. 3DCS thus provides the first rigorous benchmark for developing physically grounded, functionally reliable 3D molecular representations. GitHub repository: https://github.com/ComDec/3DCS.

## 1 Introduction

Molecular representations (MRs) underpin a wide range of design tasks in drug discovery and catalysis (Zhang et al., 2025a;b; Yu et al., 2025a). Among these, representations derived from 3D conformations are critical (Musaelian et al., 2023; Liao & Smidt, 2022; Fang et al., 2022) because the properties of molecules and the efficacy of drugs are mainly governed by their 3D structures (Aldeghi et al., 2014; Morley et al., 2013; Chen et al., 2025; Peng et al., 2025). In drug design, for example, the success of a ligand often depends on whether its 3D conformation binds precisely and tightly to the target protein (Sousa et al., 2006; Boehr et al., 2009; Nicklaus et al., 1995).

Early molecular representations, including hand-crafted and data-driven learned, focus on 1D sequences (SMILES (Weininger, 1988)) or 2D graphs. Well-known examples include SMILES-BERT (Wang et al., 2019), Extended-Connectivity Fingerprints (ECFP) (Rogers & Hahn, 2010), and MolCLR (Wang et al., 2022). Although effective in many settings, these approaches cannot distinguish between different 3D conformers because they neglect atomic coordinates (Yu et al., 2025a). In recent years, research has expanded toward 3D MRs. Manual descriptors such as E3FP (Axen et al., 2017) extend 2D circular fingerprints into three dimensions by encoding neighboring atoms within concentric shells. Data-driven MRs include autoencoder models and transformers that encode atom positions (Zhou et al., 2023) or voxelized 3D grids (Lu et al., 2025a). Models inspired by machine-learning force fields (MLFFs) (Unke et al., 2021), such as GemNet, learn energy and force mappings from spherical harmonics (Gasteiger et al., 2024).

Although 3D MRs have advanced, the lack of a corresponding benchmark severely limits our ability to measure progress. Existing molecular benchmarks typically regard molecules as static entities and evaluate representations on tasks such as property regression or cross-molecule classification (Zhong et al., 2024; Guo et al., 2023). For example, MoleculeNet (Wu et al., 2018) aggregated

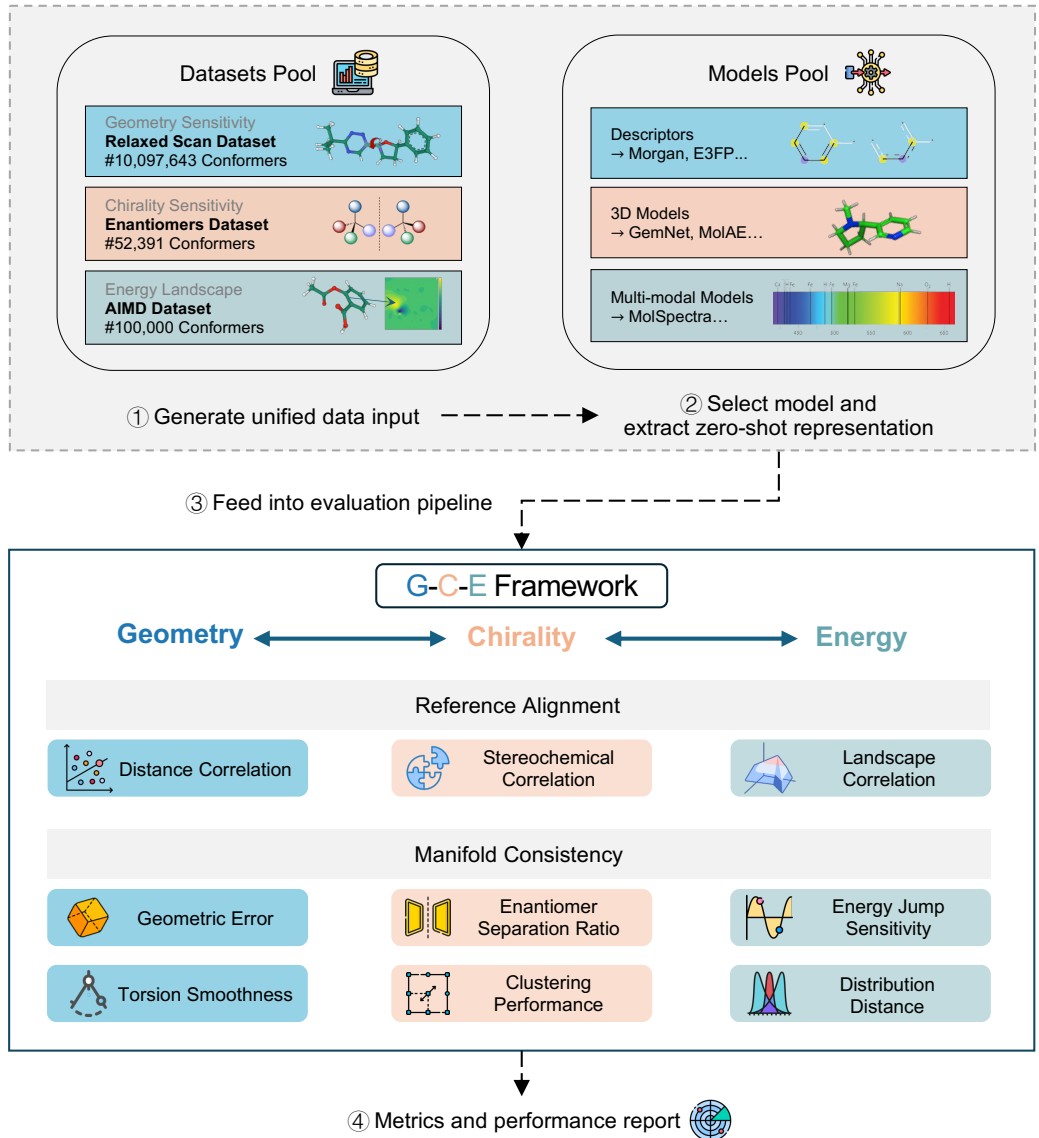

Figure 1: Overview of the 3DCS benchmark workflow. Three curated datasets (relaxed scans, enantiomers, and AIMD trajectories) are unified and processed through different representation types (descriptor, 3D, and multi-modal). Model's representations are evaluated under the G–C–E framework, which includes three dimensions: geometry, chirality, and energy with tailored metrics.

datasets from quantum chemistry and physiology, but overlooked conformational data. Subsequent benchmarks began incorporating conformers but continued to emphasize inter-molecular tasks. For example, Molecule3D (Xu et al., 2021) focuses on predicting 3D geometry from 2D graphs, while recent MARCEL (Zhu et al., 2023) leverages multiple conformers to improve molecular property prediction. We describe these as inter-molecular tasks because they exploit conformer information across different molecules, yet they do not evaluate whether representations can discriminate between conformers of the same molecule. However, many real-world applications are inherently intra-molecular conformer-sensitive: distinct conformers of the same molecule may bind a protein pocket with different affinities (Morris & Lim-Wilby, 2008) or show various quantum properties (Zhang et al., 2018). To date, no benchmark systematically evaluates how well MRs capture conformational differences within a molecule.

To address this gap, we introduce 3DCS, a new dataset and benchmark designed to systematically evaluate the sensitivity of intra-molecular 3D MRs with respect to geometry, chirality, and the conformational energy landscape. Geometric sensitivity requires that representations capture variations in distances, bond angles, and torsions, rather than collapsing distinct 3D shapes into indistinguishable representations (Gillespie & Popelier, 2001; Atz et al., 2021). Chirality sensitivity is crucial for differentiating enantiomers, which may exhibit vastly different biological activities but are often overlooked or mishandled by both handcrafted descriptors and machine learning models (Adams et al., 2021; H. Brooks et al., 2011; Peng et al., 2025). The energy sensitivity reflects the fact that the Boltzmann weight of a conformer governs its probability of occurrence; representations that do not align with relative energies risk mischaracterizing the significance of the conformer (Cercignani, 1988). This work makes three main contributions:

- Large-scale datasets for conformer sensitivity. We curate three datasets that contain over 1M molecules and 11M conformers: (i) a relaxed scan dataset with $\sim$1.5M molecules and $\sim$10 million conformers obtained by rotating an inter-ring bond and relaxing the rest of the structure; (ii) a chirality dataset with $\sim$4K chiral drug-like molecules from ChEMBL where enantiomers and enantiomers are enumerated, yielding $\sim$52K conformers; and (iii) an energy dataset derived from Ab-initio Molecular Dynamics (AIMD) that includes high-quality energies and forces computed with density functional theory.

- We propose a unified evaluation framework named GCE with two layers: (i) reference alignment tests whether representation distances align with physical references such as RMSD, dihedral angles, chirality signatures, and energy differences; (ii) manifold consistency assesses whether representations form coherent manifolds, preserving neighborhoods, separating enantiomers, and reflecting energy jumps. The framework is model-agnostic and applies to both handcrafted descriptors and learned representations.

- We evaluate classical descriptors (e.g., E3FP) and a broad range of learned representations (UniMol, MolAE, MolSpectra, GemNet, MACE, and the field-based encoder FMG) in a zero-shot setting, and additionally study their behavior under supervised fine-tuning. Results show that while learned models effectively capture geometric variations, they inconsistently handle chirality and poorly align with the energy landscape. These findings highlight critical gaps and offer actionable guidance for designing physically faithful 3D molecular representations. In additional fine-tuning experiments, we further show that 3DCS chirality and energy metrics remain informative after training on 3DCS itself.

## 2 RELATED WORK

**Molecular benchmarks.** Early benchmarks, such as MoleculeNet (Wu et al., 2018), collate datasets across quantum chemistry, physiology, and biophysics, but treat molecules as single static graphs or structures. Subsequent benchmarks began to probe conformational robustness: Molecule3D (Xu et al., 2021) evaluates geometry prediction from 2D graphs, while Zhang & Vitalis (2025) introduces datasets for benchmarking classification robustness under different conformers. MARCEL (Zhu et al., 2023) introduces datasets to learn over conformer ensembles for property prediction, and GEOM (Axelrod & Gomez-Bombarelli, 2022) provides energy-annotated conformations for generative models (Lu et al., 2025b). While these efforts mark progress, they still focus on inter-molecular tasks (e.g., property regression or classification across molecules) and do not directly test how well a representation distinguishes conformers of the same molecule. Our 3DCS benchmark explicitly shifts the focus to intra-molecular conformational sensitivity.

**Molecular representations.** MRs can be derived either from handcrafted descriptors that encode chemical domain knowledge or data-driven representation learning. Molecular fingerprints such as ECFP/Morgan (Rogers & Hahn, 2010), MACCS keys (Durant et al., 2002), and E3FP (Axen et al., 2017) are intended to represent the presence (or absence) of substructures within molecules, often in a binary vector. They are explainable and remain strong baselines, but their sparse nature contributes to their insensitivity in capturing complex chemistries, such as global geometry or potential energy (Wigh et al., 2022; Venkatraman et al., 2024).

In contrast, data-driven approaches learn representations directly from raw molecular modalities, which can broadly be grouped into three categories: 1D sequences, 2D molecular graphs, and 3D structures. The most widely adopted 1D representation is SMILES, explored by models such as

seq2seq (Xu et al., 2017) and SMILES-BERT (Wang et al., 2019). At the 2D level, models such as MolCLR (Wang et al., 2022), MolGNet (Li et al., 2021), and GROVER (Rong et al., 2020) used graph neural networks. Hybrid strategies further enrich 2D encoders with conformational information, e.g., 3D-Infomax (Stärk et al., 2022) and GraphMVP (Liu et al., 2021). Recently, 3D-based MRs have achieved state-of-the-art performance on property benchmarks. Some, such as UniMol (Zhou et al., 2023) and MolAE (Yang et al., 2024), use Cartesian coordinates directly, while others take pre-defined equivariant basis functions (e.g., spherical harmonics or SO(3) tensor features) to encode geometry (Gasteiger et al., 2024; Fuchs et al., 2020). There are also multi-modal approaches. For example, MolLM (Tang et al., 2024; Zhao et al., 2024) integrates 2D biomedical text with 3D molecular structures, while MolSpectra (Wang et al., 2025) combines spectral data with 3D information to enhance representation learning. Despite these advances, most models are trained on a single low-energy conformer, meaning they may not recognize the diversity of a conformer ensemble.

## 3 DATASETS

We design the 3D Conformational Sensitivity (3DCS) benchmark based on three well-curated datasets, each targeting a different aspect of conformational representation: geometric flexibility, chirality, and potential energy. All molecules and conformers used in 3DCS will be released together with data-generation scripts and evaluation code in our public GitHub repository.

**Geometry Dataset: Relaxed Scans.**  To probe MR's ability to capture geometric changes within the same molecule, we curated a large-scale dataset of organic molecules. The dataset is generated through the following steps (Chen et al., 2025).

1. Scaffold generation: connect heterocyclic units into diverse scaffolds;
2. Substituent diversification: introduce two specialized substituent libraries, randomly attaching them to scaffolds, yielding 1,559,779 unique molecules;
3. Relaxed scan: for each molecule, perform a relaxed dihedral scan around the inter-ring single bond in $2.5°$ increments, using `xTB` with the best `crude` precision level for geometric optimization, recording the single point energies. The constraint is done on-the-fly with `force constant=0.05` parameter;
4. Redundancy reduction: apply Density-Based Spatial Clustering of Applications with Noise (DBSCAN) clustering with `eps=0.5` to remove redundant conformers for each molecule.

After processing, the dataset comprises 10,097,643 conformers, each annotated with its corresponding dihedral and energy values. Implementation details can be found in Appendix C.1. We use xTB to perform the relaxed scans, which strikes a practical balance between accuracy and scale for our $\sim$10M-conformer corpus (Yu et al., 2025b). Recent systematic studies have shown that GFN-type xTB methods reproduce DFT-optimized geometries of organic and $\pi$-conjugated molecules with high fidelity (e.g., heavy-atom RMSD and bond/angle statistics with $R^2 \geq 0.99$).

**Chirality Dataset: Chiral Drug Candidates.**  To evaluate model sensitivity to chirality, we curate drug-like molecules with annotated stereocenters from ChEMBL (Gaulton et al., 2012). We filter drug-candidate molecules with chirality annotations, yielding 4,057 diverse molecules and conduct geometry optimization on each stereoisomer with `xTB`.

We retain only pairs that pass stereochemical validation (R/S consistency) after optimization, ensuring that observed representation differences reflect chirality rather than failures in geometry optimization. This protocol yields 15,218 conformers across enantiomer pairs. We apply small torsional perturbations within each stereoisomer so that conformer ensembles of different enantiomers overlap in geometry space, preventing models from separating enantiomers using coarse geometric differences alone instead of stereochemistry.

After perturbation and GFN-xTB optimization, we recompute the R/S configuration of every stereocenter and discard any conformer whose assignment changes, yielding 52,391 perturbed conformers that strictly preserve the intended stereochemistry for enantiomer separation.

For supervised chirality experiments, we use an 8:1:1 Murcko scaffold split into train/validation/test sets to avoid trivial leakage across enantiomers with similar substructures.

**Energy Dataset: AIMD trajectories.** For evaluating energy and force fidelity, we adopt the Revised MD17 dataset (Christensen & Von Lilienfeld, 2020), which contains high-quality molecular dynamics trajectories of 10 small organic molecules. Each trajectory provides 100K conformers sampled along MD simulations, annotated with DFT-level energies and atomic forces. This benchmark assesses whether molecular representations remain consistent with the underlying potential energy surface (PES).

## 4 GEOMETRY–CHIRALITY–ENERGY (GCE) EVALUATION FRAMEWORK

Our goal is to quantify how well a MR captures conformational variability. The **GCE** framework consists of two layers: *reference alignment* and *manifold consistency*. Given a molecule $S$ with conformers $C = \{c_1, c_2, ...c_n\}$, we have the following metrics.

### 4.1 REFERENCE ALIGNMENT

We compare pairwise representations distances $\Delta_{ij}$ (See Appendix C.2) with three physically motivated references:

1. **Geometry.** The reference geometry distance is

$$D_{ij}^{(G)} = \text{RMSD}(c_i, c_j),$$

i.e., the Root–Mean–Square Deviation (RMSD) of atom positions after optimal alignment. Throughout this work, RMSD values are reported in Ångström (Å). Geometry-sensitive representations should preserve the relative ordering of $D^{(G)}$.

2. **Chirality.** For a molecule with $m$ stereocenters, each conformer $c_i$ has a chirality signature

$$\chi(c_i) = (s_1, \ldots, s_m) \in \{\pm 1\}^m.$$

Where $+1/-1$ represents relative configurations R/S or D/L. The chirality distance is the fraction of mismatched stereocenters:

$$D_{ij}^{(C)} = \tfrac{1}{m} \sum_{t=1}^{m} \mathbf{1}\big[s_t(c_i) \neq s_t(c_j)\big].$$

A chirality-sensitive representation should place enantiomers far apart.

3. **Energy.** The energy distance reflects differences in potential energy:

$$D_{ij}^{(E)} = |E(c_i) - E(c_j)|.$$

Energy sensitive representations should scale with $D^{(E)}$ and highlight large energy jumps. Energies in the AIMD energy dataset are reported in kcal/mol, while energies in the relaxed-scan dataset are stored in *Hartree*.

We compute $\Delta_{ij}$ as cosine distance for learned representations and Tanimoto distance for binary fingerprints. Alignment is quantified using: Spearman and Kendall rank correlations, RBF kernel centered kernel alignment (CKA) (Kornblith et al., 2019), and Isotonic regression $R^2$ (monotonic fidelity). High correlations indicate that representation distances respect physical references.

### 4.2 MANIFOLD CONSISTENCY

Beyond pairwise alignment, we test whether representations form a coherent manifold. Details for metrics can be found in Appendix C.

1. **Local Isometry Error (LIE).** For each conformer, we compare neighborhoods under $D^{(G)}$ and $\Delta$, computing RMSD of distances normalized by local averages. Low LIE indicates preservation of local geometric neighborhoods.

2. **Torsion correlation and smoothness.** For the geometry dataset, we correlate angular (dihedral) distances with $\Delta$. We also compute angular smoothness (AS), the median representation change per degree of dihedral rotation. Moderate AS values indicate responsive yet stable representations. See details in Appendix C.4.

3. **Chirality separation.** We evaluate enantiomers separation via: Enantiomers Separation AUC (ESA–AUC), Nearest–neighbour accuracy (NN1–acc), and Silhouette coefficients (with molecule configurations labels). We also compute Hopkins statistics and the best–$k$ silhouette to detect latent clusters without adding labels. See details in Appendix C.5.

4. **Energy jump sensitivity (EJS) and smoothness.** We measure the conditional probability that large energy differences imply large representation separations, and compute thresholded smoothness of representation trajectories relative to energy changes. Thresholded smoothness focuses on segments with significant energy jumps. Additionally, we report Kolmogorov–Smirnov distances between the distributions of representation distances and energy differences. See details in Appendix C.6.

## 5 EXPERIMENTS AND RESULTS

Since conformers of the same molecule share identical 1D and 2D structures, we restrict our evaluation to representations that explicitly encode 3D information. Our benchmark evaluates both handcrafted and learned representations under a common framework.

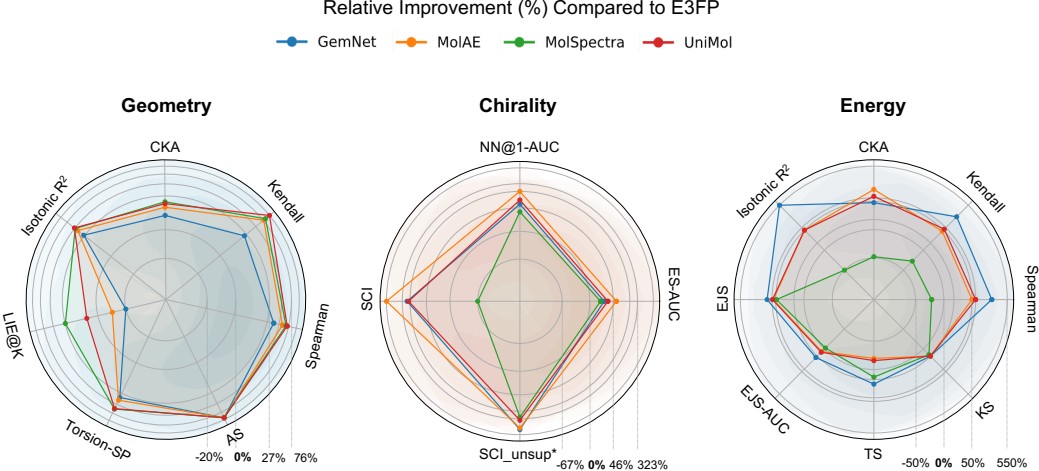

Figure 2: Radar chart summarizing the benchmark results across GCE dimensions. Each axis corresponds to one metric within its domain, and values are relative improvements over the E3FP baseline. SCI_unsup∗: Silhouette coefficients without configuration labels.

**Baseline Overview.** We evaluate classical 3D fingerprints, field-based encoders, and recent 3D molecular encoders that differ in input modality, geometric inductive bias, and training objectives.

**E3FP**. E3FP (Axen et al., 2017) constructs alignment-free 3D manual molecular fingerprints by iteratively drawing shells around each atom and hashing the local 3D substructures—including non-bonded neighbors and stereochemistry—into a folded, fixed-length bit vector. Key parameters include the shell radius multiplier, the number of iterations, whether to include stereochemistry, and the final bit length.

**UniMol**. The UniMol framework employs an SE(3)-equivariant Transformer with rotation-invariant positional encoding and pairwise atomic representations (Zhou et al., 2023). Pretraining comprises (i) masked atom prediction—masking and recovering atom types—and (ii) a 3D coordinate denoising objective that adds Gaussian noise to a fraction of coordinates and trains the model to recover the noise-perturbed positions and atom types via equivariant heads.

**MolAE**. MolAE (Yang et al., 2024) introduces an encoder–decoder architecture with a 3D Cloze Test objective to address two issues in encoder-only coordinate-denoising frameworks: objective mismatch between pretraining and downstream tasks, and potential disruption of atom identity caused by coordinate noise. Instead of corrupting coordinates with noise, it drops a subset of atoms

and their positions, encodes the remaining structure, and uses a decoder with positional encoding to reconstruct the full coordinates and atom types. This preserves the encoder representation and avoids positional bias.

**GemNet**. GemNet (Gasteiger et al., 2024) is a geometric message-passing network that predicts energies and forces by encoding distances, angles, and dihedral angles through direction-aware two-hop message passing. It utilizes spherical Fourier–Bessel bases (including radial, circular, and spherical components) and symmetric filters to capture geometric features, and it trains with a weighted loss on energy and force to learn both the potential and its gradient. Although originally designed for property prediction, its spherical harmonics feature and architectural bias toward rich geometric cues tend to produce informative molecular embeddings.

**MolSpectra**. MolSpectra (Wang et al., 2025) augments 3D denoising with quantum-mechanical spectrum information via SpecFormer, a patch-based Transformer that jointly encodes UV–Vis, IR, and Raman spectra by dividing each spectrum into patches, projecting them with learnable positional encoding, and concatenating them across modalities. Pretraining combines 3D coordinate denoising, masked patch reconstruction on spectral patches, and cross-modal contrastive learning to align spectral and geometric representations, leading to 3D encoders that outperform denoising-only baselines.

**MACE**. MACE (Batatia et al., 2023) is a E(3)-equivariant architecture for atomistic modeling that combines message passing with spherical harmonic features to predict energies and forces with high sample efficiency. Similar to GemNet, it is trained with explicit energy/force supervision on MLFF-style datasets, but it uses a different parameterization of equivariant tensors and interaction blocks. We evaluate MACE as a strong energy–force baseline on our benchmark.

**FMG**. FMG (Dumitrescu et al., 2025) is a field-based generative model that represents each molecule as multi-channel 3D voxel grids of atom and bond density fields, obtained by placing Gaussian radial basis functions on atom centers and bond midpoints and evaluating them on a fixed lattice. To avoid E(3)-equivariance while still sharing statistical strength across conformations, FMG aligns each molecule to a canonical frame via PCA-based reference rotations, thereby sacrificing rotational invariance but retaining sensitivity to reflections and hence chirality. We treat FMG as a strong, chiral-aware field baseline that contrasts our point-cloud encoders by explicitly trading exact rotational symmetry for richer geometric and stereochemical fidelity.

## 5.1 Geometry Experiments and Results

We observe from Table 1 and Figure 2 that nearly all learned representations outperform hand-crafted molecular fingerprints. Notably, models directly using atomic coordinates outperform GemNet with spherical harmonics. We speculate that this is because spherical harmonics are composed of relative distances, but necessarily sacrifice some absolute geometric information when we use the RMSD of atom positions as a reference.

At the AS metric level, molecular fingerprints yield much larger values than learned representations. This arises because fingerprints encode the presence or absence of substructure as binary vectors: even minor dihedral changes can flip multiple bits, leading to disproportionately large pairwise distances, despite the underlying conformational shift being small. In contrast, learned representations distribute structural variation more smoothly across the molecule, dampening local perturbations and producing smaller AS values.

For GemNet, relative spherical harmonics are employed, so torsional changes manifest only as perturbations in a limited set of distance and angular descriptors. In contrast, coordinate-based models translate torsional rotations into broader positional shifts across multiple atoms, resulting in higher AS values compared to GemNet. Nevertheless, models that incorporate additional chemical information—such as SMILES strings or energy spectra—avoid excessive sensitivity to minor conformational changes. This reflects a desirable form of robustness: they capture stereochemical variation while remaining stable against slight conformational fluctuations.

Table 1: Geometry benchmark metrics comparing learned representations and handcrafted fingerprints. The number of local neighborhoods for LIE@k is 3 by default.

| Metric | E3FP | GemNet | MolAE | MolSpectra | UniMol |
|---|---|---|---|---|---|
| Spearman ($\uparrow$) | 0.406 | 0.560 | 0.640 | 0.682 | **0.697** |
| Kendall ($\uparrow$) | 0.272 | 0.336 | 0.483 | 0.506 | **0.563** |
| CKA ($\uparrow$) | 0.757 | 0.813 | 0.862 | **0.904** | 0.889 |
| Isotonic $R^2$ ($\uparrow$) | 0.521 | 0.667 | 0.734 | 0.770 | **0.782** |
| LIE@k ($\downarrow$) | 0.324 | 0.390 | 0.349 | **0.238** | 0.306 |
| Torsion-SP ($\uparrow$) | 0.313 | 0.420 | 0.437 | **0.508** | 0.504 |
| AS | 2.757 | 0.0018 | 0.0048 | 0.0214 | 0.006 |

## 5.2 CHIRALITY EXPERIMENTS AND RESULTS

We find that nearly all learned representations perform poorly against the chirality reference when evaluated with rank-based correlation metrics (see Appendix B.1). This result is unsurprising: the chirality reference is inherently discrete, and virtual chirality descriptors such as OPD (Osipov et al., 1995) are ill-suited for direct correlation analysis. It is not meaningful to assume that conformers with more stereocenter mismatches should necessarily lie farther apart in representation space. Rather, the key question is whether representations can reliably distinguish enantiomers, independent of their absolute correlation with discrete chirality distances.

For this reason, in Table 2 we focus exclusively on chirality separation results. Here, we find that MolAE achieves the best performance. Notably, both MolAE and UniMol adopt the same SE(3)-invariant framework; the key difference lies in their training objectives. UniMol is trained to denoise perturbed conformers by reconstructing the original coordinates. However, random perturbations can easily induce chirality inversion, effectively misaligning the training signal (e.g., the model may learn to map an R conformer back to an S conformer). In contrast, MolAE employs a 3D cloze task, where parts of the molecule—including potential stereocenters—are masked and must be reconstructed. Because enantiomers differ subtly around stereogenic centers, this objective naturally encourages the model to capture those differences and learn a faithful mapping. With the addition of the field-based FMG encoder, we observe that chirality-aware architectures can indeed exploit the structure of our benchmark. FMG matches or exceeds MolAE in ES–AUC despite not being pre-trained on large molecular datasets.

**Chirality fine-tuning.** Further, on the 3DCS–chirality dataset, we train a supervised contrastive loss on encoder's embeddings using Murcko scaffold splits. For each model, we encourage conformers of the same stereoisomer to cluster tightly while pushing apart conformers of different stereoisomers of the same parent molecule. Fine-tuning improves absolute chirality metrics across all models (see Table 5), but the relative ranking largely persists: MolAE and UniMol remain strong among SE(3)-equivariant encoders, GemNet and MACE improve yet still lag behind the best chirality-aware representations. Notably, GemNet (we use GemNet-Q) benefits from its extra dihedral features and outperforms MACE on chiral discrimination. The field-based FMG encoder achieves the best overall performance, with ES–AUC approaching 0.999.

Table 2: Chirality benchmark. The listed metrics focus on the ability of models to distinguish between stereochemical configurations. Fine-tuned baselines results are reported in Appendix table 5.

| Metric | E3FP | GemNet | MolAE | MolSpectra | UniMol | FMG | MACE |
|---|---|---|---|---|---|---|---|
| ES–AUC ($\uparrow$) | 0.486 | 0.577 | **0.782** | 0.545 | 0.622 | 0.706 | 0.485 |
| NN@1–Acc ($\uparrow$) | 0.178 | 0.292 | **0.497** | 0.235 | 0.339 | 0.412 | 0.199 |
| Hopkins ($\uparrow$) | – | 0.593 | 0.602 | 0.533 | 0.559 | **0.752** | 0.637 |
| SCI ($\uparrow$) | -0.012 | 0.015 | 0.115 | -0.020 | 0.012 | **0.117** | -0.094 |
| $\text{SCI}_{unsup}$*($\uparrow$) | 0.033 | 0.272 | 0.247 | 0.127 | 0.152 | **0.509** | 0.369 |

5.3 ENERGY EXPERIMENTS AND RESULTS

As shown in Table 3, GemNet and MACE deliver the strongest performance on our energy-based metrics, particularly in the energy jump detection task, reflecting their training objectives that explicitly incorporate energy and force information. Their spherical-harmonic inputs are constructed from relative distances, which naturally map to potential energy, likely enhancing their ability to align with energetic variation. In contrast, for most models, correlations between representations and energy differences remain much weaker than those with geometric distances, suggesting that molecular representations generally encode geometric variation while carrying little information about energetic variation if they haven't trained on energy or force targets.

By contrast, molecular fingerprint E3FP shows substantially poor correspondence with energy. The reason is that counts of substructure are energy-unsensitive, so distances in fingerprint space are hard to reflect energetic contrast. Notably, if we restrict attention to segments of significant energy shifts (our threshold smoothness metric), E3FP's performance improves. At more coarse-grained scales, it retains the capacity to recognize structural changes with significant energy differences.

**Energy prediction fine-tuning.** On the Revised MD17 dataset, we attach a small MLP head to each encoder and train it to predict energies on the official train/test splits. Since RMD17 does not provide a validation set, we use a fixed training schedule and report the performance of the last checkpoint on the held-out test set, avoiding tuning on test data. For the fixed E3FP fingerprint, we train a linear head only. We find that models with better 3DCS energy scores (e.g., MACE, GemNet) consistently achieve lower energy MAE after fine-tuning, and the method ranking induced by 3DCS energy metrics matches the downstream ranking. Detailed results for fine-tuning experiments are reported in Appendix Table 8- 9.

Additionally, we test whether learned representations recover structure in PES. We sample conformers from molecular trajectories and select the dihedral angles with the largest variance as collective variables (CVs) . For the representations, we perform PCA and use the first two principal components as proxy CVs. Surprisingly, the resulting 2D representations often reveal energy barriers and minima, suggesting that these learned representations could be useful for analyzing transition states and reaction paths (Appendix A.4).

Table 3: Energy benchmark measuring correspondence between representation distances and energetic variation.

| Metric | E3FP | GemNet | MolAE | MolSpectra | UniMol | FMG | MACE |
|---|---|---|---|---|---|---|---|
| Spearman (↑) | 0.026 | 0.078 | 0.039 | 0.023 | 0.043 | 0.015 | **0.236** |
| Kendall (↑) | 0.018 | 0.052 | 0.026 | 0.015 | 0.028 | 0.009 | **0.159** |
| CKA (↑) | 0.011 | 0.016 | **0.024** | 0.006 | 0.019 | 0.011 | 0.017 |
| iso $R^2$ (↑) | 0.002 | 0.013 | 0.003 | 0.001 | 0.003 | 0.001 | **0.080** |
| EJS (↑) | 0.184 | 0.356 | 0.294 | 0.271 | 0.301 | 0.269 | **0.578** |
| EJS-ROCAUC (↑) | 0.531 | 0.592 | 0.545 | 0.526 | 0.549 | 0.517 | **0.764** |
| TS | 0.385 | **0.444** | 0.346 | 0.409 | 0.356 | 0.582 | 0.424 |
| KS (↑) | 0.916 | 0.998 | 0.998 | 0.977 | 0.997 | 0.999 | **0.999** |

6 DISCUSSION

Our benchmarks show that most molecular representations behave as compressed encodings of a molecule's geometric degrees of freedom—atomic coordinates, bond angles, and torsions. SE(3)-equivariant transformers (Fuchs et al., 2020) enforce rigid-motion symmetry and can propagate information across the whole molecule, whereas standard message-passing networks primarily capture local neighborhoods and exhibit a strong locality bias. Sparse fingerprints such as E3FP focus on local substructures and miss many aspects of global geometry.

Across all baselines, enantiomer separation is uniformly weak in the zero-shot setting. This is expected: most models are pre-trained without explicit stereochemical supervision, and coordinate-denoising or reconstruction objectives can inadvertently encourage mappings that flip chirality. Our

chirality benchmark, together with the field-based FMG encoder, makes this failure mode explicit. FMG, which is designed to overcome equivariant limitations on chirality, achieves substantially better enantiomer clustering than conventional SE(3)-equivariant encoders and reaches near-perfect ES–AUC after supervised contrastive fine-tuning on 3DCS–chirality.

Energy representation remains a major open challenge. For most encoders, correlations between representation distances and conformer energies are consistently much weaker than those with geometric distances: representations tend to track geometry but only weakly reflect the underlying potential energy surface. GemNet and MACE are notable exceptions: by combining spherical-harmonic inputs with explicit energy/force supervision, they exhibit much stronger energy alignment and better sensitivity to large energy jumps. Our downstream RMD17 experiments further show that models with higher 3DCS energy scores also attain lower energy prediction error after fine-tuning, indicating that 3DCS energy metrics are predictive of downstream energy-learning difficulty rather than purely diagnostic.

Overall, 3DCS provides a starting point and fundamental tool for studying molecules at the conformer-ensemble level. Placing a new model on our geometry–chirality–energy radar immediately highlights whether its weaknesses lie in geometric smoothness, enantiomer separation, or energy alignment, and thus where to invest modeling effort or hyperparameter tuning. In this work we focus on unconditional, ligand-side conformational competence; extending 3DCS to conditional settings such as protein–ligand docking, pose prediction, and chirality-sensitive property prediction is a natural next step, and our public datasets and code are designed to make such extensions and cross-correlations with downstream benchmarks straightforward.

**Design recommendations.** These findings point to three directions for future benchmarks and models: (i) enrich geometry-sensitive objectives beyond Cartesian recovery to include bond angles and torsions; (ii) dataset construction and loss functions should explicitly account for stereochemical integrity. Chirality-preserving augmentations and enantiomers-aware objectives are critical; (iii) using spherical harmonic bases rather than raw coordinates can reduce redundancy and degrees of freedom, making the representation more compatible with energetic features.

## 7 CONCLUSION

3DCS focuses molecular representation evaluation around *intra-molecular* conformational sensitivity. Through three large datasets and the unified GCE framework, we reveal that modern MRs excel at capturing geometry but lag on chirality and energy. By identifying these gaps, 3DCS provides concrete directions for developing the next generation of molecular encoders that jointly honor geometry, chirality, and energy. We aim for this benchmark to catalyze community-wide efforts in developing pretraining objectives and datasets that more faithfully reflect the dynamic 3D nature of molecules.

## 8 THE USAGE OF LANGUAGE MODELS

We used an LLM to improve paper writing (grammar and wording); all authors reviewed the edits and take full responsibility for the content.

## 9 REPRODUCIBILITY STATEMENT

To promote reproducibility, we release all the original datasets and the scripts used to reproduce baselines. Code for running the experiment and charts generation is also provided. Additionally, step-by-step instructions, code examples, and corresponding files are included in the GitHub repository (https://github.com/ComDec/3DCS).

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

# A SUPPLEMENTARY FIGURES

In this section, we provide additional figures and commentary to elucidate the constitution of the 3DCS benchmark.

## A.1 STATISTIC OF DATASETS

The **geometry dataset** summarizes fundamental molecular descriptors and energy annotations for the conformers used in the benchmark. Conformer energies are computed using the semi-empirical GFN2-xTB method within the xTB software. Figure 3 presents the distribution of these descriptors and the xTB energy spectrum for a representative subset of conformers.

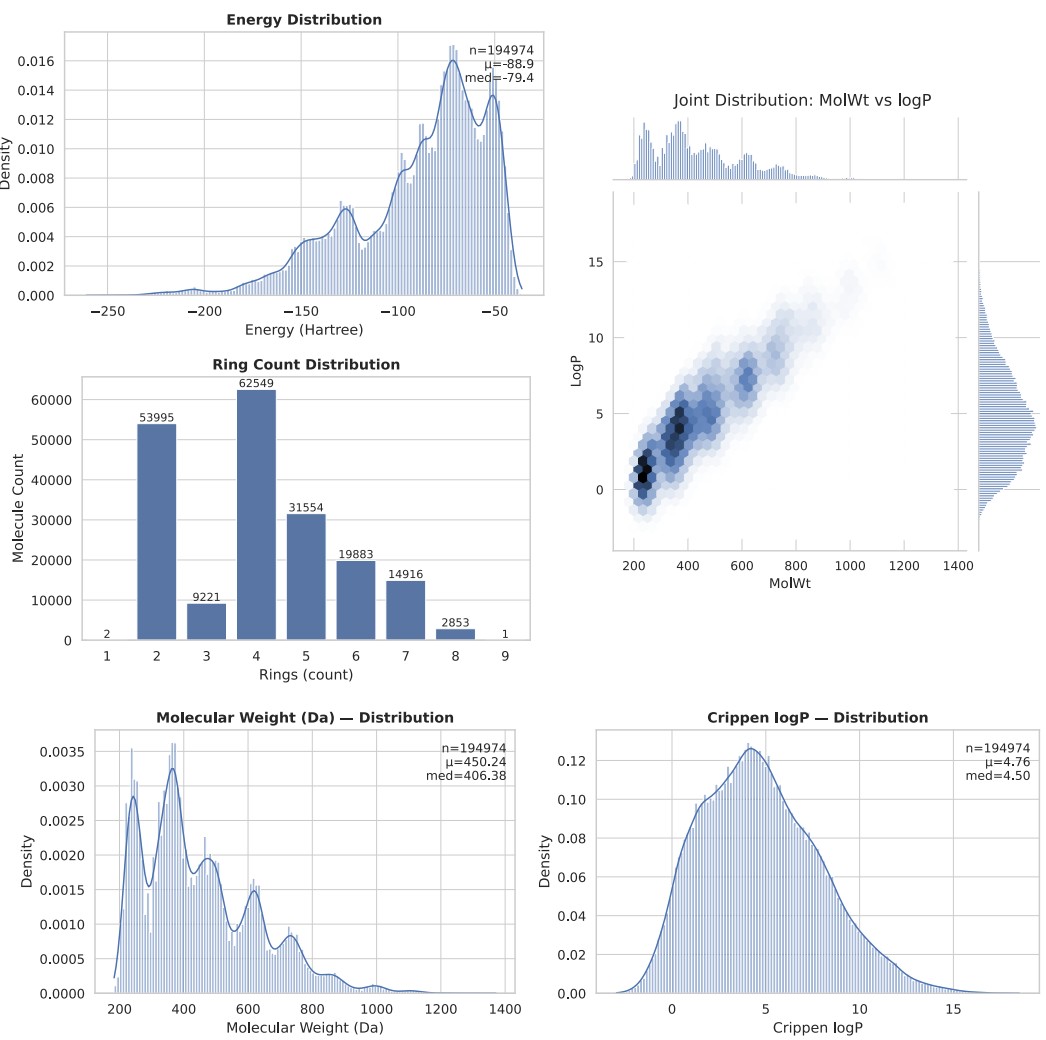

Figure 3: Summary of molecular and energetic descriptors for the geometry dataset.

The **chirality dataset** focuses on stereochemical sensitivity. It includes molecules with one or more stereocenters and records whether each stereocenter has a defined R or S configuration. Figure 4 summarizes these general stereochemistry-related statistics.

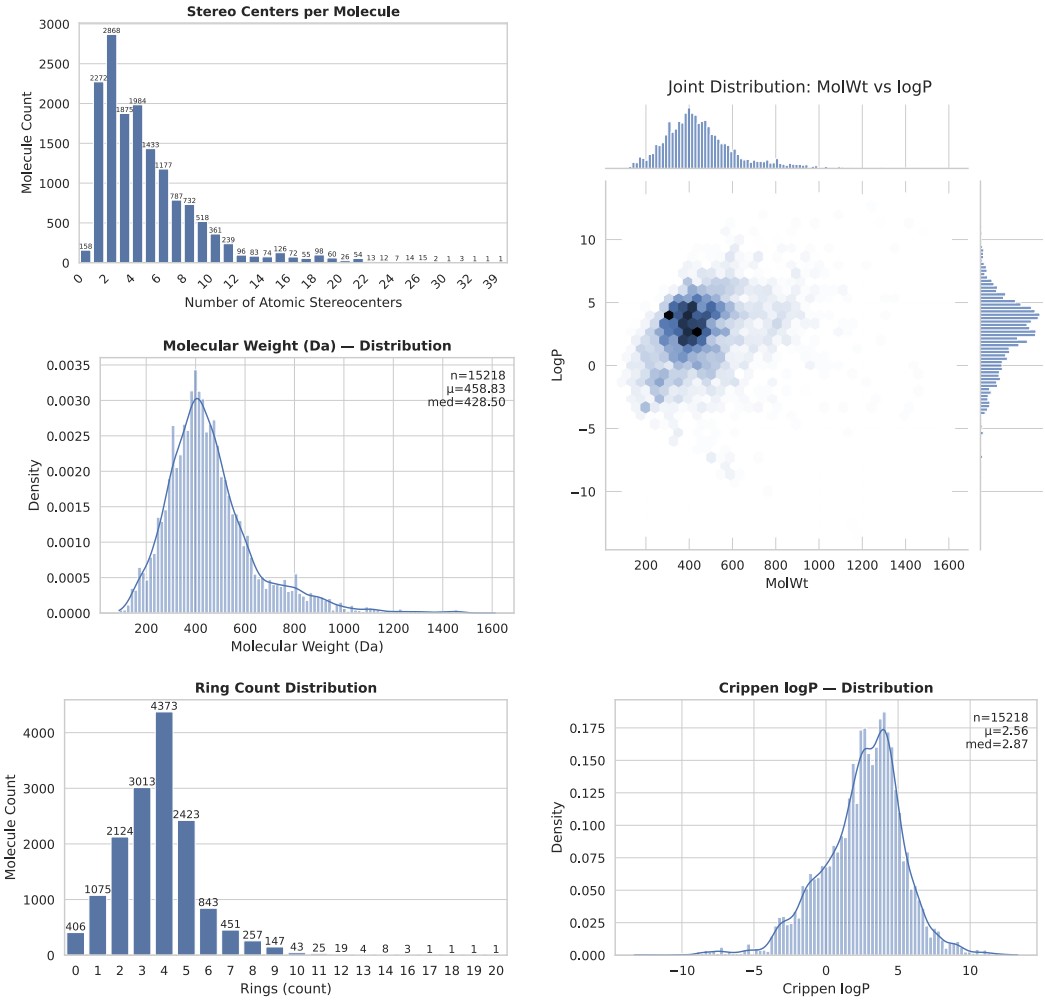

Figure 4: Statistics of the chirality dataset.

## A.2 VIOLIN CHARTS OF GEOMETRY BENCHMARKS

We present violin plots that summarize the distribution of geometry benchmark scores across baseline models. Each violin depicts the kernel density of per-molecule scores, allowing us to visualize both the central tendency and the spread of results, as well as potential asymmetries in the distribution. To assess whether the observed differences between models are statistically significant, we conducted pairwise comparisons using the two-sided Mann–Whitney $U$ test on the per-molecule score distributions. See Figure 5.

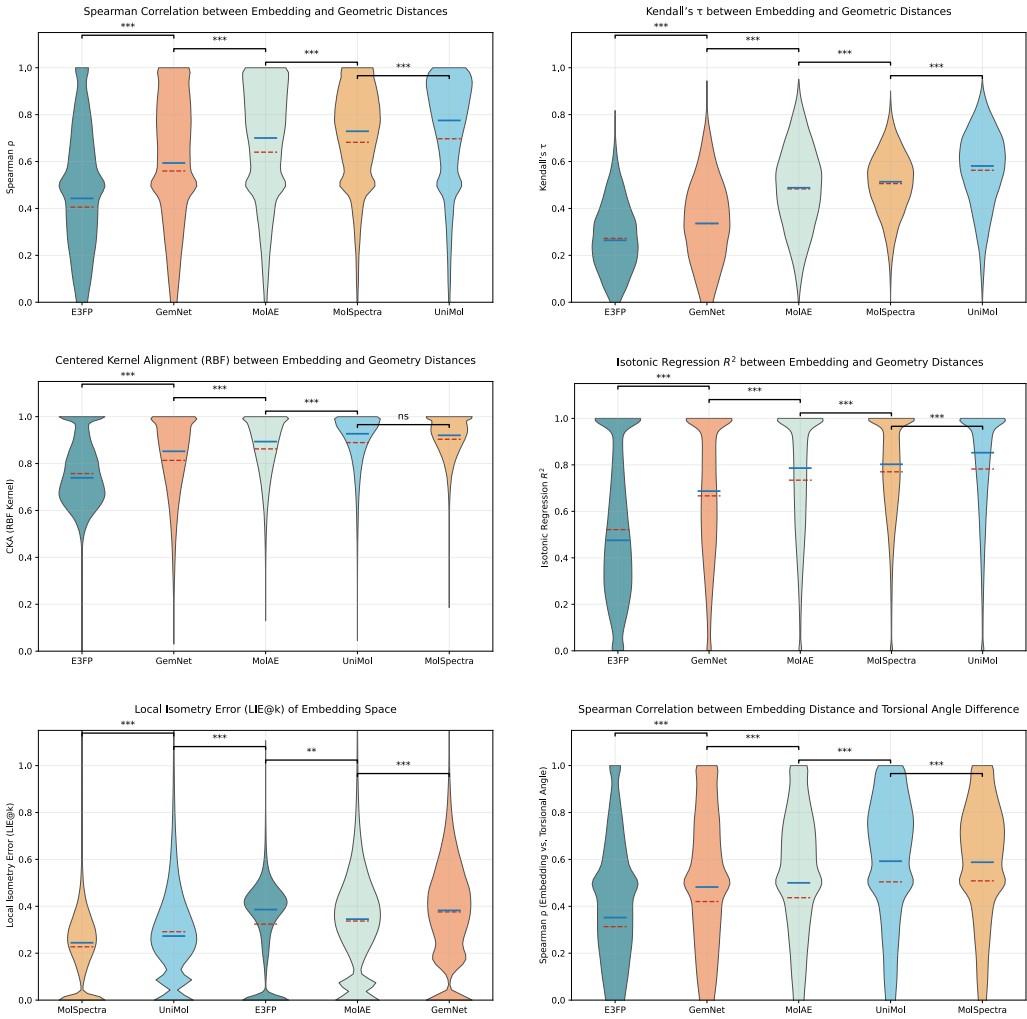

Figure 5: Violin plots summarizing the geometry benchmark results across baseline models. Pairwise statistical comparisons were conducted using the Mann–Whitney $U$ test. Significance levels are annotated as follows: "ns" for $p \geq 0.05$, "*" for $p < 0.05$, "**" for $p < 0.01$, and "***" for $p < 0.001$.

## A.3 Violin Charts of Chirality Benchmarks

We visualize the per-molecule outcomes for chirality-sensitive metrics using violin plots. Similarly, we perform two-sided Mann–Whitney $U$ tests on the per-molecule chirality scores. See Figure 6. DBI: Davies-Bouldin Index

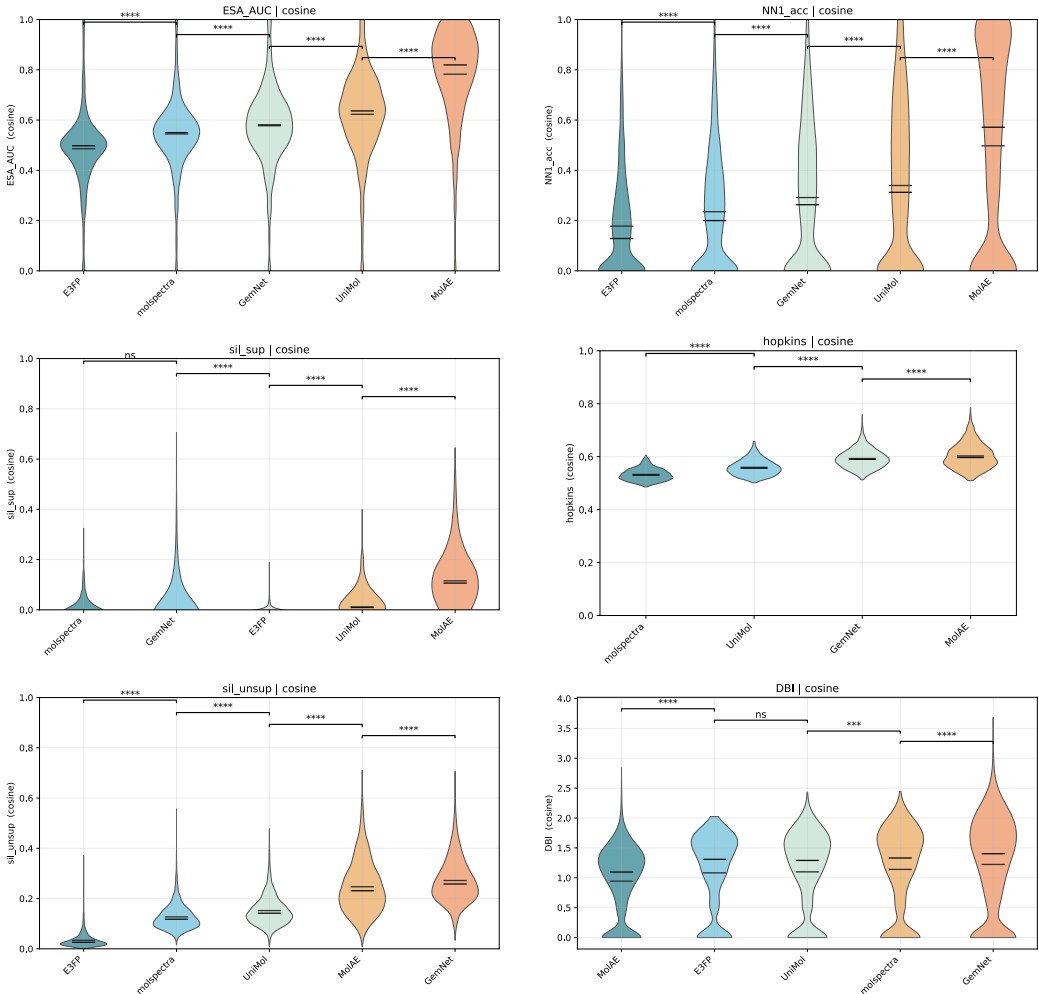

Figure 6: Violin plots summarizing the chirality benchmark results across baseline models. Pairwise statistical comparisons were conducted using the two-sided Mann–Whitney $U$ test on per-molecule score distributions.

## A.4 PES FROM SAMPLED TRAJECTORIES

A potential energy surface (PES) maps molecular energy across conformations. Since the full PES is high-dimensional, we visualize 2D versions: reference PES from the two most variant dihedral angles, and representation-based PES from PCA projections of learned embeddings. If geometry–energy relations are preserved, characteristic features—peaks, valleys, and saddles—should remain visible in the representation space (Fig. 7).

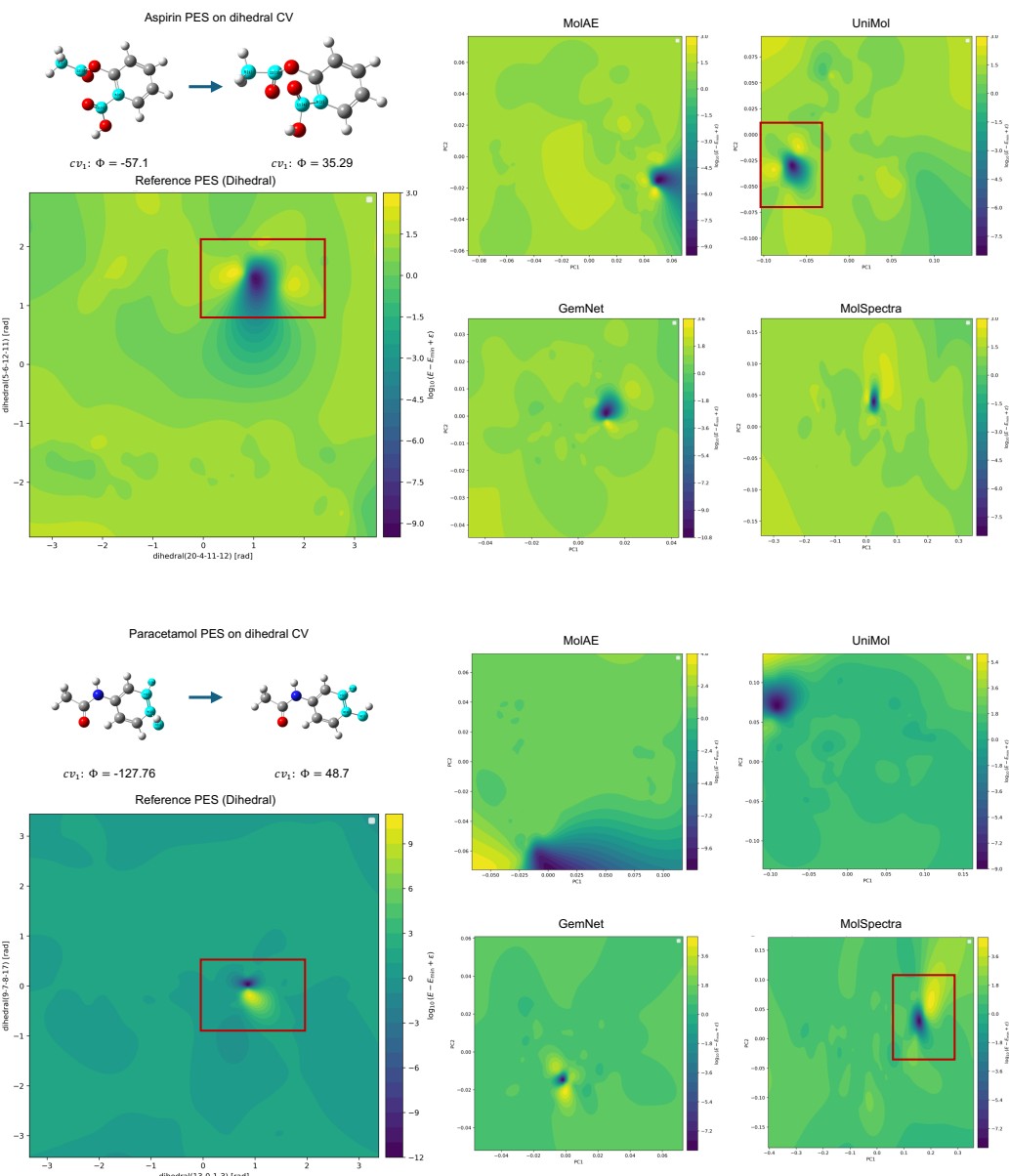

Figure 7: Potential energy surfaces (PES) comparison for aspirin and paracetamol. Left panels show ground-truth PES as a function of the two dihedral angles with the highest variance; right panels show the PES in representation space (learned representations projected via PCA). Red boxes mark energy peaks (barrier regions) that are partly captured by representations, illustrating which structural/energetic features are preserved versus which are lost or smoothed out.

# B  SUPPLEMENTARY TABLES

## B.1  SUPPLEMENTARY CHIRALITY BENCHMARK

Table 4: Chirality metrics of correlation between representation distance and chirality reference, including stereochemical labels and OPD index.

| Metric | E3FP | GemNet | MolAE | MolSpectra | UniMol |
|---|---|---|---|---|---|
| Spearman | 0.092 | **0.009** | -0.012 | -0.000 | -0.025 |
| Kendall | **0.078** | 0.006 | -0.012 | -0.003 | -0.022 |
| CKA | **0.789** | 0.657 | 0.647 | 0.715 | 0.659 |
| OPD–Spearman | 0.011 | -0.015 | **0.006** | -0.008 | 0.008 |
| OPD–dCor | **0.833** | 0.707 | 0.729 | 0.793 | 0.749 |
| OPD–iso $R^2$ | **0.256** | 0.240 | 0.251 | 0.244 | 0.255 |

Table 5: Fine-tuned model performance on chirality datasets. We use supervised contrastive learning to train the models. E3FP is not a trainable model. The result align well with our zero-shot experiment and agreement to model architecture, while E(3) network like GemNet and MACE cannot represent chirality well.

| Metric | ES-AUC |
|---|---|
| E3FP | N/A |
| UniMol | 0.976 |
| MolAE | 0.971 |
| MolSpectra | 0.987 |
| FMG | **0.999** |
| GemNet | 0.9001 |
| MACE | 0.887 |

## B.2  SUPPLEMENTARY ENERGY BENCHMARK

Table 6: Summary of representations' performance on the energy dataset. Values are reported as mean $\pm$ 95% CI.

| Metric | E3FP | GemNet | MolAE | MolSpectra | MACE | FMG |
|---|---|---|---|---|---|---|
| Spearman | $0.026 \pm 0.001$ | $0.078 \pm 0.002$ | $0.039 \pm 0.001$ | $0.023 \pm 0.001$ | $\mathbf{0.236 \pm 0.011}$ | $0.015 \pm 0.004$ |
| Kendall | $0.018 \pm 0.001$ | $0.052 \pm 0.002$ | $0.026 \pm 0.001$ | $0.015 \pm 0.001$ | $\mathbf{0.160 \pm 0.008}$ | $0.010 \pm 0.003$ |
| CKA | $0.011 \pm 0.000$ | $0.016 \pm 0.000$ | $\mathbf{0.024 \pm 0.000}$ | $0.006 \pm 0.000$ | $0.017 \pm 0.001$ | $0.012 \pm 0.001$ |
| iso $R^2$ | $0.002 \pm 0.000$ | $0.013 \pm 0.001$ | $0.003 \pm 0.000$ | $0.001 \pm 0.000$ | $\mathbf{0.083 \pm 0.010}$ | $0.001 \pm 0.000$ |
| EJS($0.1\sigma$) | $0.151 \pm 0.006$ | $0.252 \pm 0.000$ | $0.251 \pm 0.000$ | $0.250 \pm 0.000$ | $\mathbf{0.255 \pm 0.000}$ | $0.250 \pm 0.000$ |
| EJS($0.5\sigma$) | $0.155 \pm 0.006$ | $0.263 \pm 0.000$ | $0.256 \pm 0.000$ | $0.253 \pm 0.000$ | $\mathbf{0.286 \pm 0.002}$ | $0.253 \pm 0.001$ |
| EJS($1\sigma$) | $0.162 \pm 0.006$ | $0.286 \pm 0.001$ | $0.265 \pm 0.001$ | $0.258 \pm 0.001$ | $\mathbf{0.350 \pm 0.007}$ | $0.257 \pm 0.002$ |
| EJS($2\sigma$) | $0.184 \pm 0.007$ | $0.356 \pm 0.004$ | $0.294 \pm 0.002$ | $0.271 \pm 0.002$ | $\mathbf{0.578 \pm 0.019}$ | $0.269 \pm 0.006$ |
| EJS($3\sigma$) | $0.217 \pm 0.009$ | $0.451 \pm 0.007$ | $0.328 \pm 0.004$ | $0.289 \pm 0.004$ | $\mathbf{0.845 \pm 0.025}$ | $0.283 \pm 0.016$ |
| ROC_AUC | $0.531 \pm 0.001$ | $0.592 \pm 0.003$ | $0.545 \pm 0.001$ | $0.527 \pm 0.001$ | $\mathbf{0.764 \pm 0.011}$ | $0.517 \pm 0.005$ |
| Smoothness | $0.847 \pm 0.002$ | $0.998 \pm 0.000$ | $0.996 \pm 0.000$ | $0.974 \pm 0.001$ | $\mathbf{0.999 \pm 0.000}$ | $0.972 \pm 0.002$ |
| TS | $0.385 \pm 0.010$ | $0.444 \pm 0.004$ | $0.346 \pm 0.001$ | $0.409 \pm 0.004$ | $0.424 \pm 0.014$ | $\mathbf{0.583 \pm 0.007}$ |
| KS | $0.913 \pm 0.001$ | $0.998 \pm 0.000$ | $0.998 \pm 0.000$ | $0.976 \pm 0.001$ | $0.999 \pm 0.000$ | $\mathbf{1.000 \pm 0.000}$ |

Table 7: Number of detected energy jumps.

| Energy Threshold | #Jumps (mean $\pm$ 95% CI) |
|---|---|
| $0.1\sigma$ | 1,839,169.111 $\pm$ 34.666 |
| $0.5\sigma$ | 1,232,341.449 $\pm$ 61.925 |
| $1\sigma$ | 638,505.706 $\pm$ 197.971 |
| $2\sigma$ | 102,542.184 $\pm$ 492.257 |
| $3\sigma$ | 8,929.385 $\pm$ 167.468 |

Table 8: DFT energy prediction MAE (kcal/mol) on the Revised MD17 dataset(part I).

| Model | Aspirin | Azobenzene | Benzene | Ethanol | Malonaldehyde |
|---|---|---|---|---|---|
| E3FP | 4.628 | 4.986 | 1.845 | 3.203 | 3.282 |
| UniMol | 5.010 | 3.509 | 0.988 | 0.797 | 0.801 |
| MolAE | 5.759 | 8.092 | 1.028 | 1.439 | 1.555 |
| MolSpectra | 3.525 | 4.173 | 0.813 | 1.554 | 1.679 |
| GemNet | 0.217 | 0.441 | 0.222 | 0.069 | 0.148 |
| FMG | 4.821 | 5.165 | 1.863 | 3.287 | 3.289 |
| MACE | **0.051** | **0.028** | **0.009** | **0.009** | **0.018** |

Table 9: DFT energy prediction MAE (kcal/mol) on the Revised MD17 dataset(part II).

| Model | Naphthalene | Paracetamol | Salicylic | Toluene | Uracil | Avgerage |
|---|---|---|---|---|---|---|
| E3FP | 4.279 | 4.871 | 4.425 | 4.009 | 3.880 | 3.941 |
| UniMol | 3.648 | 2.814 | 6.491 | 9.289 | 2.922 | 3.627 |
| MolAE | 2.098 | 2.814 | 5.609 | 4.080 | 5.570 | 3.804 |
| MolSpectra | 2.868 | 3.600 | 3.160 | 2.683 | 2.462 | 2.652 |
| GemNet | 0.249 | 1.312 | 0.930 | 0.017 | 0.106 | 0.371 |
| FMG | 4.550 | 5.009 | 4.509 | 4.015 | 4.054 | 4.056 |
| MACE | **0.012** | **0.030** | **0.021** | **0.012** | **0.012** | **0.020** |

## C IMPLEMENTATION DETAILS FOR DATASETS AND METRICS

### C.1 GEOMETRY DATASET: PROGRAMMATIC CONSTRUCTION DETAILS

**Pipeline recap.** We build *bi-aryl/heteroaryl* backbones and decorate them with a merged library of ring scaffolds and small functional groups. RDKit is used for fragment connection, sanitization, and 3D seeding via ETKDG. xTB software is used for geometric optimization and relaxed scan. Candidate linkage atoms are selected among non-saturated heavy atoms while reserving one position per pair for the final inter-ring bond. To control quality, we cap extra substitution sites per fragment ($\leq 2$) and stratify ring choices by molecular weight (heavy/light extremes + random mid-slice).

**Libraries.** During generation, we *pool* ring scaffolds and functional groups into a single substituent library; any entry can serve as a backbone unit or as an R-group donor depending on context. A minimal illustrative subset is shown in Table 10; the **complete, machine-readable lists and scripts** are in our GitHub repository.

Table 10: Illustrative subset of the merged library used for geometry datasets construction.

| Class | SMILES |
|---|---|
| Functional group | C(=O)O |
| Functional group | CO |
| Functional group | CC(=O)C |
| Functional group | C=O |
| Functional group | COC |
| Functional group | C#N |
| Functional group | C=NC |
| Functional group | CCl |
| Bulky subunit | C1=CC=CC=C1 |
| Bulky subunit | C12=CC=CC=C1C=CC=C2 |
| Bulky subunit | C12=CC=CC=C1C=C3C(C=CC=C3)=C2 |
| Bulky subunit | C1(PC2=CC=CC=C2)=CC=CC=C1 |
| Bulky subunit | CC(C)C |
| Ring/scaffold | C1CNCC1 |
| Ring/scaffold | C1N=CC=C1 |
| Ring/scaffold | C1=COC=C1 |
| Ring/scaffold | C1=CSC=C1 |
| Ring/scaffold | O1CCCCC1 |
| Ring/scaffold | N1=CC=CC=N1 |
| Ring/scaffold | S1CCNCC1 |
| Ring/scaffold | C1=CC=C2CC=CC2=C1 |
| Ring/scaffold | C1=CC=C2C=COC2=C1 |
| Ring/scaffold | C1=CC=C2OC(=O)C=CC2=C1 |

## C.2 REPRESENTATIONS DISTANCES

For handcrafted descriptors, molecular fingerprints $F_i = f(c_i)$ are compared using Tanimoto distance. For learned representations, we extract zero-shot representations $z_i = g(c_i)$ without fine-tuning. The representation distance $\Delta_{ij}$ is defined as:

$$\Delta_{ij} = \begin{cases} 1 - \dfrac{z_i^\top z_j}{\|z_i\|\|z_j\|}, & \text{Cosine distance} \\ \|z_i - z_j\|_2, & \text{Euclidean distance} \\ 1 - \dfrac{|F_i \cap F_j|}{|F_i \cup F_j|}, & \text{Tanimoto distance} \end{cases}$$

Cosine distance is the default for learned representations, as most encoders implicitly normalize their representations. Tanimoto is reserved for binary fingerprints, where overlap naturally defines similarity. All the distance matrices are normalized to the 0-1 range.

## C.3 CORRELATION METRICS

**Rank correlations.**

$$\rho_S(m) = \text{Spearman}\big(\text{vec}_\triangle(D), \ \text{vec}_\triangle(\Delta)\big), \tag{1}$$

$$\tau(m) = \text{Kendall}\big(\text{vec}_\triangle(D), \ \text{vec}_\triangle(\Delta)\big). \tag{2}$$

**Centered Kernel Alignment (CKA).** Define RBF kernels

$$K_{ij}^{(D)} = \exp\Big(-\frac{(D_{ij})^2}{2\sigma_D^2}\Big), \quad K_{ij}^{(\Delta)} = \exp\Big(-\frac{\Delta_{ij}^2}{2\sigma_\Delta^2}\Big),$$

with bandwidths by the median heuristic on upper-triangular entries. Let $H = I - \frac{1}{n_m}\mathbf{1}\mathbf{1}^\top$ and $\tilde{K} = HKH$ be centered kernels. Then

$$\text{CKA} = \frac{\langle \tilde{K}^{(D)}, \tilde{K}^{(\Delta)} \rangle_F}{\|\tilde{K}^{(D)}\|_F \, \|\tilde{K}^{(\Delta)}\|_F}.$$

**Isotonic Regression $R^2$.** We perform an isotonic regression fit:

$$\hat{D}_{ij} = f_{\text{iso}}(\Delta_{ij}), \qquad f_{\text{iso}} \text{ monotone non-decreasing,}$$

mapping representation distances $\Delta_{ij}$ onto the reference values $D_{ij}$. We then compute the coefficient of determination.

$$\text{isoR}^2 = 1 - \frac{\sum_{i<j}(D_{ij} - \hat{D}_{ij})^2}{\sum_{i<j}(D_{ij} - \bar{D})^2}.$$

A high isoR$^2$ score indicates that the representation preserves the reference ordering well enough that a monotone transformation recovers the reference scale.

### C.4 GEOMETRY METRICS

We use $D_{ij}^{(G)}$, the RMSD matrix between conformers.

**Local Isometry Error (LIE).** Global correlations can hide distortions at the neighborhood level. To test neighborhood preservation, for each conformer $i$ we identify its $k$ nearest neighbors $\mathcal{N}_k^{(D)}(i)$ under $D^{(G)}$ and compute

$$\text{LIE}_i = \sqrt{\frac{1}{k} \sum_{j \in \mathcal{N}_k^{(D)}(i)} \left( \frac{D_{ij}^{(G)}}{\bar{D}_i} - \frac{\Delta_{ij}}{\bar{\Delta}_i} \right)^2},$$

with $\bar{D}_i, \bar{\Delta}_i$ denoting local averages. The final score is the mean over conformers.

**Torsion correlation and smoothness.** Beyond RMSD, a relaxed scan is performed on a controlled axis of conformational change. For torsional scans along a dihedral angle $\phi_i$, we define circular angular distance:

$$A_{ij} = \min\big(|\phi_i - \phi_j|, \, 2\pi - |\phi_i - \phi_j|\big).$$

The torsion–representation rank correlation is defined as:

$$\rho_{\text{tor}} = \text{corr}_S\big(\text{vec}(A), \, \text{vec}(\Delta)\big)$$

The torsion correlation tests global alignment with torsional change.

To further quantify continuity, we define angular smoothness (AS) by sorting conformers along the scan and measuring displacement per angular increment:

$$s_i = \frac{\|z_{i+1} - z_i\|}{|\phi_{i+1} - \phi_i|}, \qquad \text{AS} = \text{median}_i s_i.$$

High $\rho_{\text{tor}}$ indicates global consistency with torsional change in conformers, while moderate AS values reflect stable local sensitivity—avoiding both insensitivity (collapse) and over-reactivity to small rotations.

### C.5 CHIRALITY METRICS

We use $D_{ij}^{(C)}$, the normalized Hamming distance between chirality signatures, as the stereochemical ground truth. For supervised enantiomers separation metrics, we use the class labels $y_i$ from the enantiomers' relative configuration tags and pairwise representation distances $\Delta_{ij}$:

**ESA-AUC (Enantiomer Separation AUC).** Treat cross-enantiomer pairs as positives and use distance as the score:

$$t_{ij} = \mathbf{1}[y_i \neq y_j], \qquad \text{ESA-AUC} = \text{AUC}\Big(\{(\Delta_{ij}, t_{ij})\}_{i<j}\Big).$$

A higher AUC indicates that between-class distances dominate within-class distances, meaning that enantiomers are well separated.

**Nearest-neighbor Accuracy.** Nearest-neighbor classification under $\Delta$:

$$\hat{j}(i) = \arg\min_{j \neq i} \Delta_{ij}, \quad \hat{y}_i = y_{\hat{j}(i)}, \quad \text{NN1-acc} = \frac{1}{n} \sum_i \mathbf{1}[\hat{y}_i = y_i].$$

This measures whether local decision boundaries align with stereochemical classes.

**Silhouette Coefficient Index.** With $a_i$ the intra-class mean distance and $b_i$ the nearest other-class mean,

$$s_i = \frac{b_i - a_i}{\max(a_i, b_i)}, \qquad \text{sil\_sup} = \frac{1}{n} \sum_i s_i.$$

Larger values indicate tighter intra-class cohesion and wider inter-class margins.

**Davies-Bouldin Index (DBI).** The Davies–Bouldin Index quantifies the trade-off between within-cluster compactness and between-cluster separation. Given a partition of the data into $K$ clusters $\{\mathcal{C}_1, \ldots, \mathcal{C}_K\}$ with centroids $\{\mu_i\}_{i=1}^K$ under a Euclidean distance metric $d(\cdot, \cdot)$, the within-cluster scatter and the inter-centroid distance are

$$S_i = \frac{1}{|\mathcal{C}_i|} \sum_{x \in \mathcal{C}_i} d(x, \mu_i), \qquad M_{ij} = d(\mu_i, \mu_j) \quad (i \neq j).$$

For each cluster $i$, define its worst-case similarity to other clusters:

$$R_{ij} = \frac{S_i + S_j}{M_{ij}}, \qquad R_i = \max_{j \neq i} R_{ij}.$$

The DBI is the average of these worst-case similarities:

$$\text{DBI} = \frac{1}{K} \sum_{i=1}^K R_i.$$

Lower values indicate better clustering: small $S_i$ (compact clusters) and large $M_{ij}$ (well-separated centroids). DBI is approximately scale-invariant with respect to linear rescalings of the distance. In practice, one may add a small $\varepsilon$ to $M_{ij}$ in the denominator to avoid degeneracy when centroids coincide.

For unsupervised metrics, we aim at evaluate the shape of representations while ignoring the chirality label. we report:

**Hopkins statistic.** A value $> 0.5$ suggests non-uniform, clusterable structure for continuous representations only.

**Best-$k$ silhouette.** We scan $k \in \{2, \ldots, k_{\max}\}$ and report

$$\text{sil\_unsup}^* = \max_k \text{silhouette}(\Delta, \text{k-means/k-medoids}_k),$$

using $k$-means for vector representations and $k$-medoids (PAM) on precomputed distances for fingerprints. A larger value indicates sharper, more coherent clusters that align with latent stereochemical organization.

## C.6 ENERGY METRICS

Let $E \in \mathbb{R}^n$ be the per-conformation energies and let $\Delta \in \mathbb{R}^{n \times n}$ denote pairwise representation distances. We write $dE_{ij} = |E_i - E_j|$ and $dZ_{ij} = \Delta_{ij}$, and evaluate pairs only on the upper triangle.

**Half-normal scale of energy differences.** We treat $\{dE_{ij}\}$ as samples from a half-normal distribution with scale $\sigma_d$ and estimate $\hat{\sigma}_d$ by:

$$\hat{\sigma}_d = \sqrt{\mathbb{E}[dE^2]}$$

**Energy-Jump Sensitivity (EJS).** For an energy-jump level $\lambda > 0$, define the energy threshold $\theta(\lambda) = \lambda\,\hat{\sigma}_d$ and the jump set $\mathcal{J}_\lambda = \{(i,j) : dE_{ij} > \theta(\lambda)\}$. Let $\tau$ be a distance threshold chosen as the 75% quantile of $dZ$. The sensitivity at level $\lambda$ is the conditional hit rate:

$$\mathrm{EJS}(\lambda) = \mathbb{E}[\mathbf{1}\{dZ_{ij} > \tau\} \mid (i,j) \in \mathcal{J}_\lambda] = \frac{1}{|\mathcal{J}_\lambda|} \sum_{(i,j) \in \mathcal{J}_\lambda} \mathbf{1}\{dZ_{ij} > \tau\}.$$

We evaluate $\lambda$ on the grid $\{0.1, 0.5, 1.0, 2.0, 3.0\}$ and report all $\mathrm{EJS}(\lambda)$ values. By default, we choose $\lambda = 2$.

**EJS–ROC (discrimination under an energy jump threshold).** Further, we turn energy-jump detection into a binary classification task with labels and scores:

$$y_{ij} = \mathbf{1}\{dE_{ij} > \theta\}, \qquad s_{ij} = dZ_{ij}.$$

In our evaluation we fix $\theta = 2\,\hat{\sigma}_d$ (i.e., $\lambda = 2.0$) to generate the binary labels. We compute $\mathrm{ROC\text{-}AUC} = \mathrm{AUC}(\{(s_{ij}, y_{ij})\})$ using `sklearn.metrics.roc_auc_score`.

**Thresholded Smoothness (TS).** Along ordered conformational trajectories, we verify whether representations vary smoothly with energy. To focus on regions of substantial energetic contrast, we restrict to segments with $|E_{\pi_{k+1}} - E_{\pi_k}| > T_E$ and compute:

$$\mathrm{TS}_{T_E} = \frac{1}{|\mathcal{K}|} \sum_{k \in \mathcal{K}} \exp\left(-\frac{\Delta_{\pi_k, \pi_{k+1}}/Q_{0.9}^{(\Delta)}}{|E_{\pi_{k+1}} - E_{\pi_k}|/Q_{0.9}^{(E)} + \varepsilon}\right).$$

This normalizes by robust (90% quantile) scales to enable comparison across molecules and penalizes representations that change abruptly where the energy landscape is smooth.

**Distributional divergence.** We ask whether the overall distribution of representation distances resembles that of energy differences. Let $F_\Delta$ and $F_E$ denote the empirical CDFs of $\mathrm{vec}(\Delta)$ and $\mathrm{vec}(D^{(E)})$, respectively. We report the two-sample Kolmogorov–Smirnov *statistic*:

$$D_{\mathrm{KS}} = \sup_x \left|F_\Delta(x) - F_E(x)\right|,$$

In our benchmark, these quantities are used as *descriptive divergences* (smaller is better).

