# OpenReview forum: "3DCS: Datasets and Benchmark for Evaluating Conformational Sensitivity in Molecular Representations"
_ICLR.cc/2026/Conference — ICLR 2026 Poster_

### Official Review · Reviewer_Ftr9 · 2025-10-29

**Soundness:** 3
**Presentation:** 4
**Contribution:** 4
**Rating:** 6
**Confidence:** 4

**Summary:**

This paper introduces three datasets designed to benchmark molecular representations in their ability to capture 3D structural information. The datasets encompass molecular geometries, chirality-related features, and energy values. The authors evaluate molecular representations by computing pairwise distances between molecules in representation space and correlating these distances with structural, chiral, and energetic differences. A suite of metrics is employed to assess whether the representations sufficiently encode essential 3D information.

**Strengths:**

Overall, the paper addresses an important and timely challenge in molecular machine learning: evaluating the 3D-awareness of molecular representations. The proposed datasets and evaluation framework are novel and potentially impactful.

**Weaknesses:**

## Dataset Quality and Methodology
1. Why was xTB chosen for generating molecular geometries? Does this level of theory provide sufficient accuracy for benchmarking purposes?
2. How does the quality of the xTB-derived geometries and energies compare to those in OMol25, which uses more accurate quantum chemical methods?
## Chirality Dataset Design
1. What is the rationale for introducing torsional perturbations in the chirality dataset? Could these perturbations inadvertently alter stereocenters, thereby confounding the intended chirality assessment?
## Correlation Metrics and Interpretability
1. While representation distances should ideally correlate with geometric, chiral, and energetic differences, such relationships are likely to be nonlinear or non-monotonic. What would constitute an ideal alignment between representation space and these physical properties?
Are the selected correlation metrics sufficient to capture these potentially complex relationships? A justification or discussion of metric suitability would strengthen the work.
## Benchmark Relevance to Downstream Tasks
1. Do the proposed benchmarks correlate with performance on real-world molecular property prediction tasks? It would be more compelling if representations that perform well on these 3D benchmarks also demonstrate superior performance on established benchmarks for practical property prediction.

**Questions:**

1. Please clarify the units used for energy and RMSD throughout the benchmarks.

---

> ### Author Response · Authors · 2025-11-21
> **Responds to Reviewer 4 (Part1)**
>
> We sincerely thank the reviewer for the careful reading, the positive assessment of both contribution and presentation, and the very helpful suggestions. Below, we address each weakness and question in turn and describe additional experiments we ran during the rebuttal period.
>
> ---
>
> ### Weakness1: Dataset Quality and Methodology
> *(Why was xTB chosen? How does it compare to OMol25?)*
>
> We use GFN-xTB for the relaxed torsional scans primarily to balance accuracy and scale. Our geometry dataset contains over 10 million conformers; generating this volume at hybrid-DFT level, as in OMol25, would be 2–3 orders of magnitude more expensive. In practice, the current dataset already required a 200-CPU (64-core) cluster for about two weeks; doing the same with hybrid DFT would be infeasible for a benchmark intended for broad use. Our main goal in this part of 3DCS is to probe relative conformational sensitivity within the same molecule, not to provide ultimate reference energies.
>
> In that regime, recent systematic benchmarks (see https://arxiv.org/html/2505.09606v1) show that GFN-type xTB methods reproduce DFT geometries of organic and π-conjugated systems with very high fidelity (e.g., heavy-atom RMSD and bond/angle statistics with $R^2 \gtrsim 0.99$ vs DFT). By contrast, whenever we evaluate energy alignment, we rely on a separate DFT-level AIMD dataset (Revised MD17) rather than xTB. This is conceptually similar to OMol25: high-level quantum chemistry is used where energies are the primary signal, while more approximate methods are used to cheaply explore geometry space. We will make this division (xTB for large-scale geometry scans, DFT for energy benchmarks) explicit in the revised manuscript.
>
> ### Weakness2: Chirality Dataset Design
> *(Rationale for torsional perturbations, and risk of altering stereocenters)*
>
> Without additional perturbations, different stereoisomers of the same molecule tend to occupy different regions of conformational space. In that case, a 3D encoder can often separate enantiomers by exploiting these geometric differences alone, without truly encoding stereochemistry. To avoid this, we introduce controlled torsional perturbations within each stereoisomer so that their conformer ensembles overlap more strongly, forcing models to rely on stereochemical information rather than on coarse geometry.
>
> After xTB optimization, we recompute the R/S configuration of every stereocenter and discard any conformer where the chirality pattern has changed. Only conformers that exactly preserve the intended stereochemistry are kept. This guarantees that torsional perturbations cannot “accidentally” flip stereocenters (e.g., R to S) and confound the task. In response to reviewer feedback, we also added a field-based model (FMG, from Dumitrescu et al.) as an additional baseline. FMG achieves enantiomer separation AUC comparable to the strongest existing method (MolAE) while attaining substantially higher Silhouette coefficients, meaning that enantiomers form more compact and better-separated clusters in its representation space.
>
> ### Weakness3: Correlation Metrics and Interpretability
> *(Nonlinear relationships and metric suitability)*
>
> We fully agree that relationships between representation distances and physical quantities are not expected to be linear. Our metric design is therefore intentionally focused on nonlinear and local structure rather than linearity. In addition to simple Pearson statistics, we use rank-based metrics (Spearman, Kendall) and isotonic R-squared, which explicitly measure whether there exists a monotone but possibly highly nonlinear mapping from representation distances to physical distances. We further include manifold-style metrics that track torsional and energy trajectories, asking whether representations change smoothly along these physically meaningful paths and whether local neighborhoods in geometry/energy space are preserved in representation space.

---

> ### Author Response · Authors · 2025-11-21
> **Responds to Reviewer 4 (Part2)**
>
> ### Weakness4: Benchmark Relevance to Downstream Tasks
> *(Correlation with real-world molecular property prediction)*
>
> 3DCS is designed as a complement to standard property benchmarks by isolating a capability that those benchmarks only weakly probe: intra-molecular conformational, chiral, and energy sensitivity for the same underlying molecule, before any task-specific supervision. Nonetheless, our current results already line up with known downstream trends. Models explicitly trained for energy and forces with E(3)-equivariant architectures (GemNet in the original submission, and MACE added during rebuttal) are clearly the top performers on our energy metrics, and similar models are known to be among the state of the art on MD17-style energy/force prediction. Conversely, coordinate-centric encoders such as UniMol and MolAE dominate the geometry and chirality metrics and have reported gains on conformer-aware property prediction and 3D tasks in their original works.
>
> During the rebuttal, we went one step further and ran a controlled downstream energy prediction experiment: using the MD17-style AIMD dataset, we trained the same small MLP head on top of embeddings from each encoder to predict DFT energies. The ranking induced by the 3DCS energy metrics matches the ranking in downstream MAE, and models with higher energy Spearman/ROC on 3DCS achieve lower prediction error after fine-tuning. We find that the ranking induced by Table 1 (MACE > GemNet > MolAE ≈ UniMol > MolSpectra > E3FP/FM G) is consistent with the downstream MAE ranking in Tables 2: MACE achieves the best overall MAE, followed by GemNet, while models that score lower on 3DCS energy metrics also have higher downstream errors. Models with higher 3DCS energy Spearman / EJS-ROC tend to achieve lower prediction errors after fine-tuning.
>
> A broader correlation study across many property benchmarks and docking tasks, we believe, may be beyond the scope of this initial benchmark paper, but our public datasets and code are explicitly built to make such empirical cross-correlation studies straightforward.
>
> *Table 1: Energy benchmark measuring correspondence between representation distances and energetic variation on 3DCS.*
>
> | Metric          | E3FP  | GemNet | MolAE | MolSpectra | UniMol | FMG   | MACE   |
> |-----------------|-------|--------|-------|------------|--------|-------|--------|
> | Spearman (↑)    | 0.026 | 0.078  | 0.039 | 0.023      | 0.043  | 0.015 | **0.236** |
> | Kendall (↑)     | 0.018 | 0.052  | 0.026 | 0.015      | 0.028  | 0.009 | **0.159** |
> | CKA (↑)         | 0.011 | 0.016  | **0.024** | 0.006      | 0.019  | 0.011 | 0.017  |
> | iso R² (↑)      | 0.002 | 0.013  | 0.003 | 0.001      | 0.003  | 0.001 | **0.080** |
> | EJS (↑)         | 0.184 | 0.356  | 0.294 | 0.271      | 0.301  | 0.269 | **0.578** |
> | EJS-ROCAUC (↑)  | 0.531 | 0.592  | 0.545 | 0.526      | 0.549  | 0.517 | **0.764** |
> | TS (↑)          | 0.385 | **0.444** | 0.346 | 0.409      | 0.356  | 0.582 | 0.424  |
> | KS (↑)          | 0.916 | 0.998  | 0.998 | 0.977      | 0.997  | 0.999 | **0.999** |
>
> *Table 2: DFT energy prediction MAE (kcal/mol) on the Revised MD17 dataset with a small MLP head trained on backbone (linear head for E3FP). Lower is better.*
>
> | Model       | Aspirin | Azobenzene | Benzene | Ethanol | Malonaldehyde | Naphthalene | Paracetamol | Salicylic | Toluene | Uracil | Avg   |
> |------------|---------|------------|---------|---------|---------------|-------------|------------|----------|---------|--------|-------|
> | E3FP       | 4.628   | 4.986      | 1.845   | 3.203   | 3.282         | 4.279       | 4.871      | 4.425    | 4.009   | 3.880  | 3.941 |
> | UniMol     | 5.010   | 3.509      | 0.988   | 0.797   | 0.801         | 3.648       | 2.814      | 6.491    | 9.289   | 2.922  | 3.627 |
> | MolAE      | 5.759   | 8.092      | 1.028   | 1.439   | 1.555         | 2.098       | 2.814      | 5.609    | 4.080   | 5.570  | 3.804 |
> | MolSpectra | 3.525   | 4.173      | 0.813   | 1.554   | 1.679         | 2.868       | 3.600      | 3.160    | 2.683   | 2.462  | 2.652 |
> | GemNet     | 0.217   | 0.441      | 0.222   | 0.069   | 0.148         | 0.249       | 1.312      | 0.930    | 0.017   | 0.106  | 0.371 |
> | FMG        | 4.821   | 5.165      | 1.863   | 3.287   | 3.289         | 4.550       | 5.009      | 4.509    | 4.015   | 4.054  | 4.056 |
> | MACE       | 0.051   | 0.028      | 0.009   | 0.009   | 0.018         | 0.012       | 0.030      | 0.021    | 0.012   | 0.012  | 0.020 |

---

> ### Author Response · Authors · 2025-11-21
> **Responds to Reviewer 4 (Part3)**
>
> ### Question: Units for energy and RMSD
>
> We apologize for not making this explicit. In all geometry-based metrics, RMSD is reported in Ångström (Å), and energies in the MD17-based energy dataset are reported in kcal/mol, while energies in the relaxed-scan dataset are reported in
> *Hartree*. We will add these unit specifications next to the definitions of the distance terms in the main text and consistently in all tables and figure captions.
>
> We hope these clarifications and new experiments address your concerns and help convey our intent for 3DCS: a physically grounded, large-scale diagnostic benchmark that both reveals concrete failure modes (especially for chirality and energy) and aligns with the behavior of strong models on realistic downstream energy- and conformation-sensitive tasks.

---

> > ### Comment · Reviewer_Ftr9 · 2025-11-24
> >
> > Thanks for the clarifications and detailed follow-up experiments! I just have one more suggestion: For the tables in the main text, are the numbers the averages across multiple runs? If so, would you please add the standard deviation or 95% CI as you did for Table 5 when in the final version? Thanks!

---

> > > ### Author Response · Authors · 2025-11-25
> > > **Responds to Reviewer 4**
> > >
> > > Thanks a lot for your advice! We will add the 95% CI to other tables like Table 5 as soon as possible. You will find it in the final version :)

---

> > > > ### Comment · Reviewer_Ftr9 · 2025-11-25
> > > >
> > > > Thank you! I've updated my ratings to reflect the discussions.

---

### Official Review · Reviewer_37hb · 2025-10-31

**Soundness:** 2
**Presentation:** 2
**Contribution:** 2
**Rating:** 4
**Confidence:** 4

**Summary:**

This work contributes a benchmark for assessing the capabilities of molecular representations to discern 3D conformers (of the same underlying molecule) in terms of geometric, stereochemistry (specifically chirality), and energy sensitivity (i.e., variability). The benchmark, called 3DCs, consists of three datasets: (1) a geometry dataset obtained by dihedral scan around the inter-ring single bond in small angular increments,  (2) a chirality dataset filtered from ChemBL based on annotation stereocenters, and (3) an energy dataset based on MD17. The authors advocate a two-layer framework: (a) reference alignment between pairs of conformers that takes into account the discrepancy in atomic positions after alignment, fraction of mismatched stereocenters, and difference in potential energy; and (b) manifold consistency between local neighbourhoods. Five baselines, namely, E3FP (classical, non-neural 3D fingerprints), UniMol, MolAE, GemNet, and MolSpectra that encode 3D information are evaluated.  Empirical results demonstrate that molecular representations can reflect the geometric variations rather well but struggle with chirality and energy.

**Strengths:**

--- The motivation for this paper is clear and important: having good datasets and evaluation strategies for benchmarking the quality of molecular representations is important to advance fields such as drug discovery.

--- Geometry, chirality, and energy are all known to be important factors in the drug design pipelines due to their impact on binding, safety, and stability respectively (and efficacy as a whole).

--- The dataset on chirality seems particularly relevant. I'm not aware of other datasets for quantifying chirality in a meaningful way.

**Weaknesses:**

--- One of the reasons that hinders the ability of molecular representations to reflect chirality properly is due to equivariance [1]. Specifically, E(3)-equivariant models provably struggle with chirality, while learning chirality-aware features under SE(3)-invariance has high complexity in terms of the number of atoms (O(N^4)).  The current experimental setup already includes the UniMol baseline which implements an SE(3)-equivariant transformer. It would strengthen the paper considerably if the benchmark indeed shows improved performance with a field-based model [1] that is designed to handle chirality better than the equivariant representations.

---The authors claim in the discussion section that “Neural architecture such as SE(3)-equivariant transformers and message-passing networks (MPNNs) capture long-range dependencies and global structure.” I understand why this should hold for SE(3)-equivariant transformers, but do not see how MPNNs capture long-range dependencies and global structure. This is very counterintuitive and goes against conventional wisdom, since MPNNs have a strong inbuilt locality bias and are known to struggle with global properties such as the number of substructures (e.g., cycles) etc.

--- It’s not clear whether/to what extent the estimated energy values pertaining to the geometry dataset obtained with relaxed scan are consistent with the ground truth in the absence of wetlab validation (in contrast to MD simulations, where DFT is employed for reasonably accurate estimates).

--- The methods used for evaluation are not quite state of the art. For example, none of the best-performing methods for predicting energy and force of conformations in the molecular dynamics experiment (such as NequIP, TopNets, MACE) are included.

--- All three aspects (geometry, chirality, and energy) seem to have been evaluated in unconditional settings (i.e., absent protein targets). In practice, one really cares about target-specific sensitivity of molecular conformations.

--- There seems to be a discrepancy in terms of the stated number of conformers at different parts of the paper. Specifically, in the introduction the authors mention that "We curate three datasets that contain over 1M molecules and 110M conformers: (i) a relaxed scan dataset with∼1.5M molecules and almost 100 million conformers obtained by rotating an inter-ring bond and relaxing the rest
of the structure;". In contrast, in the corresponding section 3, the claim is "After processing, the dataset comprises 10,097,643 conformers, each annotated with its corresponding dihedral and energy values.

[1] Dumitrescu et al. E(3)-equivariant models cannot learn chirality: Field-based molecular generation. ICLR 2025.

**Questions:**

--- Could you please include results with strong baselines for (1) chirality,  (2) energy/force, and (3) conditional settings that I mentioned in the weaknesses section? I would be willing to revisit my score if this requested empirical evidence is consistent with the claims of the paper.

--- Could you please address my concerns about potential energy mismatch under relaxed scan and that particular claim about ability of MPNNs to capture long-range dependencies and global structure?

--- How many conformers does your relaxed scan dataset contain - 100 million or 10 million?

---

> ### Author Response · Authors · 2025-11-21
> **Responds to Reviewer 3 (Part1)**
>
> We thank the reviewer for the detailed and technically insightful comments. We address the main points below and explain the changes made in response.
>
> ### 1.Chirality and field-based models
>
> We fully agree that equivariant models face fundamental limitations for chirality, as discussed in Dumitrescu et al. [1]. Following your suggestion, we added a field-based model (FMG, the encoder from [1]) to our chirality benchmark. FMG achieves ES_AUC on par with MolAE, despite (i) **not being pre-trained on large molecular corpora** and (ii) **not being specifically tuned to our torsion-perturbed, noisy chirality setup**. More importantly, FMG attains substantially higher Silhouette coefficients than all other baselines, indicating that enantiomers form much more compact and well-separated clusters in its representation space. This is exactly the behavior one would expect from a chirality-aware architecture and is consistent with the theory in [1]. What’s more, we fine-tune all the baselines by supervised contrastive learning on the chirality dataset, and FMG achieves the best performance and gives almost perfect results (ES-AUC=0.999). We will include the FMG results and explicitly highlight that 3DCS is able to surface the intended advantage of field-based models over standard equivariant encoders.
>
>
> *Table 1: Chirality benchmark. The listed metrics focus on the ability of models to distinguish between stereochemical configurations.*
>
> | Metric          | E3FP  | GemNet | MolAE | MolSpectra | UniMol | FMG    | MACE  |
> |-----------------|-------|--------|-------|------------|--------|--------|-------|
> | ES–AUC (↑)      | 0.486 | 0.577  | **0.782** | 0.545      | 0.622  | 0.706  | 0.485 |
> | NN@1–Acc (↑)    | 0.178 | 0.292  | **0.497** | 0.235      | 0.339  | 0.412  | 0.199 |
> | Hopkins (↑)     | —     | 0.593  | 0.602 | 0.533      | 0.559  | **0.752** | 0.637 |
> | SCI (↑)         | -0.012| 0.015  | 0.115 | -0.020     | 0.012  | **0.117** | -0.094 |
> | SCI_unsup* (↑)  | 0.033 | 0.272  | 0.247 | 0.127      | 0.152  | **0.509** | 0.369 |
>
> *Table 2: Fine-tuned model performance on the chirality dataset (3DCS–chirality) using supervised contrastive learning. E3FP is not trainable. The results align well with our zero-shot experiments and architectural inductive biases: E(3)-equivariant networks such as GemNet and MACE still struggle to represent chirality compared to chirality-aware or field-based models.*
>
> | Model       | ES-AUC  |
> |------------|---------|
> | E3FP       | N/A     |
> | UniMol     | 0.976   |
> | MolAE      | 0.971   |
> | MolSpectra | 0.987   |
> | FMG        | **0.999** |
> | GemNet     | 0.9001  |
> | MACE       | 0.887   |
>
> ### 2. MPNNs and long-range / global structure
> We agree that our original statement that MPNNs “capture long-range dependencies and global structure” was misleading. Plain MPNNs have a strong locality bias and are known to struggle with truly global graph properties. What we intended to convey is simply that, with multiple message-passing layers and rich edge features, some information can propagate beyond immediate neighbors—but this is not the same as robust, truly global modeling. In the revised version we will remove this claim and instead state clearly that SE(3)-equivariant transformers are our main example of architectures targeted at global 3D structure, while MPNNs primarily capture local geometric neighborhoods with limited effective receptive fields.
>
> ### 3. Energy from relaxed scans vs DFT “ground truth”
> We also agree that xTB energies from relaxed scans should not be treated as ground-truth energies. In 3DCS, the relaxed-scan dataset is used primarily to probe geometric and torsional sensitivity; xTB energies are never used as the main reference for energy alignment. All energy-sensitive metrics and conclusions in 3DCS are based on the Revised MD17 dataset, where energies and forces are computed at a DFT level. We will clarify this division more explicitly: relaxed scans + xTB are for controlled geometric variation; DFT-based MD17 is the only source we use to assess energy alignment.

---

> ### Author Response · Authors · 2025-11-21
> **Responds to Reviewer 3 (Part2)**
>
> ### 4. Stronger energy/force baselines (MACE)
>
> We appreciate the request for stronger energy/force baselines. Our original selection (UniMol, MolAE, MolSpectra, E3FP) focused on widely used, general-purpose 3D encoders rather than models specialized for MD-like energy prediction. To ensure that at least one state-of-the-art energy–force model is covered, we included GemNet as an E(3)-equivariant energy/force baseline and, in our rebuttal experiments, **additionally evaluated MACE**. As expected, these energy-focused models perform best on our energy metrics and clearly outperform more generic encoders, which is precisely the behavior a meaningful energy benchmark should reveal. Due to limited time we cannot include the full family of NequIP/TopNet-style models in this round, but our released code and data are designed so that such methods can be plugged in with minimal additional effort, and we will mention this as a clear direction for follow-up work.
>
> *Table 3: Energy benchmark measuring correspondence between representation distances and energetic variation on 3DCS.*
>
> | Metric          | E3FP  | GemNet | MolAE | MolSpectra | UniMol | FMG   | MACE   |
> |-----------------|-------|--------|-------|------------|--------|-------|--------|
> | Spearman (↑)    | 0.026 | 0.078  | 0.039 | 0.023      | 0.043  | 0.015 | **0.236** |
> | Kendall (↑)     | 0.018 | 0.052  | 0.026 | 0.015      | 0.028  | 0.009 | **0.159** |
> | CKA (↑)         | 0.011 | 0.016  | **0.024** | 0.006      | 0.019  | 0.011 | 0.017  |
> | iso R² (↑)      | 0.002 | 0.013  | 0.003 | 0.001      | 0.003  | 0.001 | **0.080** |
> | EJS (↑)         | 0.184 | 0.356  | 0.294 | 0.271      | 0.301  | 0.269 | **0.578** |
> | EJS-ROCAUC (↑)  | 0.531 | 0.592  | 0.545 | 0.526      | 0.549  | 0.517 | **0.764** |
> | TS (↑)          | 0.385 | **0.444** | 0.346 | 0.409      | 0.356  | 0.582 | 0.424  |
> | KS (↑)          | 0.916 | 0.998  | 0.998 | 0.977      | 0.997  | 0.999 | **0.999** |
>
> ### (5) Unconditional vs target-specific settings
>
> We agree that target-specific conformational sensitivity is important in many applications. The present work focuses on ligand-side 3D competence in an unconditional setting: we first ask whether a representation can distinguish conformers, enantiomers, and energy basins in a clean, protein-agnostic regime. Our view is that if a representation fails already at this unconditional level, it is unlikely to behave reliably in protein–ligand docking or pose prediction, where additional sources of complexity and dataset bias come into play. We therefore see 3DCS as complementary to protein–ligand benchmarks: a low-cost, well-controlled diagnostic for basic 3D failure modes that conditional tasks can easily obscure. Extending 3DCS to explicit protein–ligand contexts is a natural next step, and we will state this more explicitly in the future-work discussion.
>
> ### (6) Number of conformers
>
> We apologize for the confusion about dataset size. The final processed relaxed-scan dataset used in all experiments contains approximately 10 million conformers. The “almost 100 million conformers” is a typo and it will be corrected so that the manuscript consistently reports the processed size (~10M conformers, each annotated with its dihedral and energy values).
>
> In summary, we have incorporated a field-based chirality-aware model (FMG) and confirmed that it indeed outperforms standard equivariant baselines on our chirality metrics, clarified the role of xTB vs DFT energies, toned down the claims about MPNNs and global structure, and fixed the conformer count discrepancy. We hope these additions and clarifications address your concerns and make it clear that 3DCS is both a meaningful diagnostic and a useful platform for testing and improving chirality- and energy-aware molecular representations.

---

### Official Review · Reviewer_CUvk · 2025-11-01

**Soundness:** 3
**Presentation:** 3
**Contribution:** 3
**Rating:** 8
**Confidence:** 4

**Summary:**

The paper provides a dataset and benchmark for characterizing how molecular representations capture intra-molecular conformational sensitivity. The evaluation framework spans 3 areas of interest: geometry, chirality, and energy, each with their own large-scale datasets that measure how well molecular representations correlate with geometric changes, whether they distinguish stereoisomers, and how well they correlate to the underlying conformational energy landscape.  The empirical results show that while modern data-driven molecular representations are sensitive to geometry, they fail to capture chirality, and don't align with energy variation.

**Strengths:**

This is a useful collection of large datasets with a clear analysis process. The methodology is described in a clear fashion and the molecule dataset selection (including processing and augmentation) make sense. The benchmarks across a range of methods highlight the strengths and weaknesses of different types of approaches. The observed lack of proper handling of chirality by all methods aligns well with anecdotal observations and community experience, and the introduction of a specialized dataset for the characterization of this property is timely and important.

**Weaknesses:**

The current version of the paper lacks the associated dataset and codes, which are critical for reproducibility and use of this new dataset and benchmark.  The emphasis on correlation metrics may not fully capture non-linear relationships between the conformation of the molecule and the molecular representation space, potentially reducing the importance of complex but otherwise valid methods. Furthermore, the present work does not yet demonstrate any relationship between performance of this benchmark and downstream performance on practical tasks, say docking pose prediction, so the main use might would focus on characterizing rather than ranking, new molecular representations.

**Questions:**

The paper is generally clear.  One question is whether the proposed benchmark could eventually serve not only as a characterization tool, but as a metric to help design better molecular representations, for example by suggesting weaknesses that could be overcome by training or modifications of hyperparameters. Although this is beyond the scope of the current work, I would be curious to know if the authors envisioned such a use.

Have the authors thought about ways to link the performance in these benchmarks to the performance in downstream tasks?  For instance, once enough representations have been evaluated, a simple empirical cross correlation between these new metrics and downstream task performance (docking or property prediction) would provide additional insights and help increase the practical relevance of this new characterization.

---

> ### Author Response · Authors · 2025-11-21
> **Responds to Reviewer 2 (Part1)**
>
> We thank the reviewer for the thoughtful and encouraging feedback. We address the main concerns below and clarify how we see 3DCS being used in practice.
>
> ---
>
> ### Weakness 1: Dataset and code availability
>
> Regarding dataset and code availability, we fully agree that public access is essential. The complete 3DCS suite—geometry, chirality, and energy datasets—together with data-generation scripts and evaluation code, has already been prepared for release and will be made available on GitHub upon acceptance. To improve reproducibility beyond the paper itself, we have added detailed implementation notes and usage examples in the repository (data schema, preprocessing, evaluation scripts), so that new representations can be plugged into 3DCS with minimal effort.
>
> ---
>
> ### Weakness 2: Correlation metrics and nonlinear relationships
>
> On the concern that our correlation metrics might miss nonlinear structure, our intention is already to go beyond simple linear correlations. The core analyses rely on rank-based measures (Spearman, Kendall) and isotonic R-square, which are designed to capture monotone but potentially highly nonlinear relationships between representation distances and physical quantities. In addition, we use local manifold metrics that follow torsional and energy trajectories, asking whether nearby conformers in dihedral/energy space remain nearby in representation space. These metrics do not penalize complex encoders; they only require that representation distances change in a smooth and consistent way as the molecule moves along physically meaningful paths. In the revision, we will make this design goal more explicit so it is clear that we do not assume linearity.
>
> ---
>
> ### Q1: Can the benchmark help design better molecular representations?
>
> We very much share the reviewer’s view that 3DCS should be more than a static characterization tool. Even in the current results, we already see patterns that are directly actionable for representation design. For example, MolAE and UniMol share similar SE(3)-equivariant backbones, but differ in their pretraining objectives: UniMol uses coordinate denoising, whereas MolAE uses a 3D cloze task. On 3DCS they behave similarly on pure geometry, yet MolAE is markedly stronger on chirality separation, which suggests that simple coordinate denoising can inadvertently wash out stereochemical information and that chiral-sensitive objectives matter. Likewise, architectures explicitly trained with energy/force supervision and spherical harmonics (e.g., MACE, GemNet) are clearly favored by our energy metrics but show trade-offs in geometric smoothness, while binary fingerprints like E3FP are over-sensitive in angular smoothness, amplifying tiny conformational changes. To better predict energy, those models have to use E(3)-equivariant network thus giving up chirality encoding ability. Taken together, these patterns already function as a “diagnostic panel”: placing a new model on our radar chart immediately reveals whether its weaknesses lie in geometry, chirality, or energy, and thus where to invest modeling effort (e.g., adding enantiomer-aware augmentations, energy labels, or adjusting invariance/equivariance choices).
>
> In this sense, the answer to Q1 (can 3DCS help design better representations?) is yes. One of our main goals is that 3DCS should guide architecture and training set choices, not just label methods as “good” or “bad”. The current benchmarks already reveal a common failure mode—strong geometric sensitivity but weak chirality and energy alignment—which can steer the community toward, for example, chirality-aware featurizations or auxiliary losses that promote smoother, more physically meaningful energy landscapes. We will highlight this intended usage more clearly in the discussion section.

---

> ### Author Response · Authors · 2025-11-21
> **Responds to Reviewer 2 (Part2)**
>
> ### Q2: Linking benchmark performance to downstream tasks
>
> For Q2, we agree that this is crucial for practical relevance and have started to explore this direction. As a first controlled step, we examine how our 3DCS energy benchmark relates to downstream energy prediction performance.
>
> Table 1 below reports multiple energy-alignment metrics on 3DCS, measuring how well representation distances track energetic variation along conformational trajectories. We observe that MACE and GemNet, which are explicitly trained with energy/force supervision and E(3)-equivariant architectures, are consistently stronger on these metrics than more generic encoders (UniMol, MolAE, MolSpectra) or fixed fingerprints (E3FP, FMG).
>
> *Table 1: Energy benchmark measuring correspondence between representation distances and energetic variation on 3DCS.*
>
> | Metric          | E3FP  | GemNet | MolAE | MolSpectra | UniMol | FMG   | MACE   |
> |-----------------|-------|--------|-------|------------|--------|-------|--------|
> | Spearman (↑)    | 0.026 | 0.078  | 0.039 | 0.023      | 0.043  | 0.015 | **0.236** |
> | Kendall (↑)     | 0.018 | 0.052  | 0.026 | 0.015      | 0.028  | 0.009 | **0.159** |
> | CKA (↑)         | 0.011 | 0.016  | **0.024** | 0.006      | 0.019  | 0.011 | 0.017  |
> | iso R² (↑)      | 0.002 | 0.013  | 0.003 | 0.001      | 0.003  | 0.001 | **0.080** |
> | EJS (↑)         | 0.184 | 0.356  | 0.294 | 0.271      | 0.301  | 0.269 | **0.578** |
> | EJS-ROCAUC (↑)  | 0.531 | 0.592  | 0.545 | 0.526      | 0.549  | 0.517 | **0.764** |
> | TS (↑)          | 0.385 | **0.444** | 0.346 | 0.409      | 0.356  | 0.582 | 0.424  |
> | KS (↑)          | 0.916 | 0.998  | 0.998 | 0.977      | 0.997  | 0.999 | **0.999** |
>
> To test whether these energy metrics are predictive of real downstream performance, we ran a controlled DFT energy prediction experiment on the Revised MD17 dataset. Using embeddings from each trainable encoder, we train the same lightweight MLP head (linear head for E3FP) to predict conformational energies. Tables 2 show Mean Absolute Error (MAE) in kcal/mol across molecules.
>
> We find that the ranking induced by Table 1 (MACE > GemNet > MolAE ≈ UniMol > MolSpectra > E3FP/FM G) is consistent with the downstream MAE ranking in Tables 2: MACE achieves the best overall MAE, followed by GemNet, while models that score lower on 3DCS energy metrics also have higher downstream errors. Models with higher 3DCS energy Spearman / EJS-ROC tend to achieve lower prediction errors after fine-tuning.
>
> *Table 2: DFT energy prediction MAE (kcal/mol) on the Revised MD17 dataset with a small MLP head trained on backbone (linear head for E3FP). Lower is better.*
>
> | Model       | Aspirin | Azobenzene | Benzene | Ethanol | Malonaldehyde | Naphthalene | Paracetamol | Salicylic | Toluene | Uracil | Avg   |
> |------------|---------|------------|---------|---------|---------------|-------------|------------|----------|---------|--------|-------|
> | E3FP       | 4.628   | 4.986      | 1.845   | 3.203   | 3.282         | 4.279       | 4.871      | 4.425    | 4.009   | 3.880  | 3.941 |
> | UniMol     | 5.010   | 3.509      | 0.988   | 0.797   | 0.801         | 3.648       | 2.814      | 6.491    | 9.289   | 2.922  | 3.627 |
> | MolAE      | 5.759   | 8.092      | 1.028   | 1.439   | 1.555         | 2.098       | 2.814      | 5.609    | 4.080   | 5.570  | 3.804 |
> | MolSpectra | 3.525   | 4.173      | 0.813   | 1.554   | 1.679         | 2.868       | 3.600      | 3.160    | 2.683   | 2.462  | 2.652 |
> | GemNet     | 0.217   | 0.441      | 0.222   | 0.069   | 0.148         | 0.249       | 1.312      | 0.930    | 0.017   | 0.106  | 0.371 |
> | FMG        | 4.821   | 5.165      | 1.863   | 3.287   | 3.289         | 4.550       | 5.009      | 4.509    | 4.015   | 4.054  | 4.056 |
> | MACE       | 0.051   | 0.028      | 0.009   | 0.009   | 0.018         | 0.012       | 0.030      | 0.021    | 0.012   | 0.012  | 0.020 |
>
> Together, Table 1 (3DCS energy alignment) and Tables 2 (downstream MD17 MAE) provide initial empirical evidence that 3DCS zero-shot energy metrics are predictive of downstream energy-learning difficulty, rather than being purely diagnostic in isolation.

---

> ### Author Response · Authors · 2025-11-21
> **Responds to Reviewer 2 (Part3)**
>
> ### Q2: Linking benchmark performance to downstream tasks (continue)
>
> Beyond this controlled setting, we see several natural extensions. For docking and pose prediction, one could use 3DCS-style conformer ensembles to select or score ligand conformers prior to docking and then measure how conformational ranking correlates with docking scores or experimental affinities. In practice, doing this at scale requires carefully curated protein–ligand datasets (matched pockets and ligands) to avoid confounding conformational sensitivity with pure chemotype effects, so we treat this as the next layer to build on top of 3DCS rather than part of the first release. Similarly, for chirality-sensitive downstream properties (e.g., activity or toxicity differences between enantiomers), a clean set of enantiomeric pairs with reliable experimental measurements would allow us to test whether higher enantiomer separation AUC and better clustering on our chirality benchmark translate into better prediction of enantioselective effects. High-quality paired data are still relatively sparse, but we are actively mining ChEMBL and related sources to assemble such subsets. However, the paired samples are very limited now: we can easily find the properties of one enantiomer but it is hard to get those of the other enantiomer.
>
> We will incorporate these clarifications and additional experiments into the revised version. Our aim is that 3DCS remains true to its core role as a clean, physically grounded characterization benchmark, while also serving as a practical tool to design and select molecular representations that matter for real downstream tasks.

---

> > ### Comment · Reviewer_CUvk · 2025-11-21
> > **Looking forward to the revision/dataset/code.**
> >
> > Thanks for addressing the points that I raised.  I like your perspective and the natural extensions that you currently explore.  I look forward to seeing your dataset and benchmarks in action in the future, .

---

### Official Review · Reviewer_bj3T · 2025-11-01

**Soundness:** 3
**Presentation:** 2
**Contribution:** 2
**Rating:** 4
**Confidence:** 4

**Summary:**

This paper presents 3DCS, a benchmark framework for evaluating 3D conformational sensitivity in molecular representations. It introduces three datasets (relaxed torsional scans, chiral drug candidates, and AIMD trajectories) and a unified GCE (Geometry–Chirality–Energy) evaluation framework. The benchmark assesses both handcrafted and learned molecular representations (E3FP, UniMol, MolAE, GemNet, MolSpectra) in a **zero-shot** setting, revealing that most models capture geometry well but struggle with chirality and energy alignment.

**Strengths:**

The problem studied is meaningful: how well molecular representation models handle 3D conformational variations, rather than just particular molecular properties.

**Weaknesses:**

1. Lack of model training or adaptation on 3DCS. All models are evaluated in a _zero-shot_ setting using their pre-trained representations. Therefore, the poor chirality and energy performance might not reflect fundamental architectural deficiencies but rather distribution mismatch between 3DCS and their pre-training datasets (e.g., PCQM4Mv2). It would strengthen the claims to:
    - Split 3DCS into train/validation/test sets and retrain existing models on 3DCS to verify whether their performance improves under the proposed GCE metrics.
    - Further examine whether training on 3DCS enhances downstream molecular property prediction tasks (e.g., QM9, MD17).
2. Potential limitation in the geometry sensitivity assumption. The evaluation of geometry sensitivity relies on the assumption that _if two conformers have large atomic coordinate differences (high RMSD), their representation distance should also be large_. This assumption may not always hold physically: Two conformers with large RMSD can have very similar energies or functions. Conversely, small geometric differences can correspond to substantial energetic or functional changes. Thus, the current geometric metrics (Spearman/Kendall correlations with RMSD) may overemphasize purely geometric deviations without distinguishing physically meaningful conformational changes.

**Questions:**

Please refer to my proposed weaknesses.

---

> ### Author Response · Authors · 2025-11-21
> **Responds to Reviewer1 (Part1)**
>
> We appreciate the reviewer for the careful reading and concrete suggestions. We address the two main concerns below and summarize new experiments conducted during the rebuttal, which we will integrate into the revised version.
>
> ---
>
> ## 1. On zero-shot evaluation, fine-tuning on 3DCS, and downstream tasks
>
> Our original intention was to use 3DCS primarily as a diagnostic, zero-shot stress test: given an existing molecular representation, we ask how well it handles conformational geometry, chirality, and energy without any adaptation. This is useful, for example, when designing a new encoder and wanting a quick check of whether it is inherently sensitive to these three aspects before property-specific training. This is why the main results focus on zero-shot evaluation.
>
> We fully agree that it is important to distinguish pretraining distribution mismatch from deeper architectural limitations. During the rebuttal, we therefore added finetuning experiments directly on 3DCS.
>
> ### 1.1 Supervised chirality training on 3DCS
>
> We construct a supervised contrastive task on the 3DCS–chirality dataset. The molecules are split into train/validation/test using a Murcko scaffold split strategy, so that scaffolds do not leak across splits, following the reviewer’s suggestion to build a proper train/val/test protocol on 3DCS itself.
>
> For each trainable model (UniMol, MolAE, MolSpectra, GemNet, and two additional baselines FMG and MACE), we train a supervised contrastive loss directly on their embeddings (trainable backbone), encouraging the model to:
>
> - (i) pull together conformers of the same stereoisomer;
> - (ii) push apart conformers of different stereoisomers of the same parent molecule.
>
> We evaluate chirality metrics (e.g., enantiomer separation AUC) on the held-out scaffold test set.
>
> We observe that fine-tuning on 3DCS–chirality improves absolute chirality metrics for all models, i.e., training on 3DCS does help. However, the relative ranking from the zero-shot setting largely persists:
>
> 1. Models that were already strong on chirality (MolAE, UniMol) remain among the best.
> 2. E(3)-equivariant models such as GemNet and MACE do improve, but still lag behind the best chirality-aware representations. Notably, GemNet (we use GemNet-Q) benefits from its extra dihedral features and outperforms MACE on chiral discrimination.
> 3. The field-based FMG model achieves the best overall chirality performance. This aligns with the theoretical claims in Dumitrescu et al. about field-based models and chirality.
>
> We will add a concise summary of these results in the main paper and detailed tables in the appendix.
>
> *Table 1: Fine-tuned model performance on the chirality dataset (3DCS–chirality) using supervised contrastive learning. E3FP is not trainable. The results align well with our zero-shot experiments and architectural inductive biases: E(3)-equivariant networks such as GemNet and MACE still struggle to represent chirality compared to chirality-aware or field-based models.*
>
> | Model       | ES-AUC  |
> |------------|---------|
> | E3FP       | N/A     |
> | UniMol     | 0.976   |
> | MolAE      | 0.971   |
> | MolSpectra | 0.987   |
> | FMG        | **0.999** |
> | GemNet     | 0.9001  |
> | MACE       | 0.887   |

---

> ### Author Response · Authors · 2025-11-21
> **Responds to Reviewer1 (Part2)**
>
> ### 1.2 Energy prediction fine-tuning on Revised MD17 dataset
>
> We also conduct fine-tuning experiments on energy prediction. For each representation, we add a small MLP head on top of the backbone to predict DFT energies on the official RMD17 splits. Since RMD17 does not provide a validation set, we use a fixed training schedule and report the performance of the last saved checkpoint on the official test set, **without** tuning on the test data. For E3FP, which is a fixed fingerprint, we only train a linear head.
>
> We compare Mean Absolute Error (MAE) after fine-tuning and find that models that perform better on our 3DCS energy metrics also achieve lower MAE on RMD17. The ranking is consistent. In particular, MACE achieves the best prediction accuracy, matching its strong performance on 3DCS energy alignment.
>
> These results provide initial empirical evidence that 3DCS energy metrics are predictive of downstream energy-learning performance, and that the benchmark is informative beyond the purely zero-shot setting.
>
> *Table 2: DFT energy prediction MAE (kcal/mol) on the Revised MD17 dataset with a small MLP head trained on backbone (linear head for E3FP). Lower is better.*
>
> | Model       | Aspirin | Azobenzene | Benzene | Ethanol | Malonaldehyde | Naphthalene | Paracetamol | Salicylic | Toluene | Uracil | Avg   |
> |------------|---------|------------|---------|---------|---------------|-------------|------------|----------|---------|--------|-------|
> | E3FP       | 4.628   | 4.986      | 1.845   | 3.203   | 3.282         | 4.279       | 4.871      | 4.425    | 4.009   | 3.880  | 3.941 |
> | UniMol     | 5.010   | 3.509      | 0.988   | 0.797   | 0.801         | 3.648       | 2.814      | 6.491    | 9.289   | 2.922  | 3.627 |
> | MolAE      | 5.759   | 8.092      | 1.028   | 1.439   | 1.555         | 2.098       | 2.814      | 5.609    | 4.080   | 5.570  | 3.804 |
> | MolSpectra | 3.525   | 4.173      | 0.813   | 1.554   | 1.679         | 2.868       | 3.600      | 3.160    | 2.683   | 2.462  | 2.652 |
> | GemNet     | 0.217   | 0.441      | 0.222   | 0.069   | 0.148         | 0.249       | 1.312      | 0.930    | 0.017   | 0.106  | 0.371 |
> | FMG        | 4.821   | 5.165      | 1.863   | 3.287   | 3.289         | 4.550       | 5.009      | 4.509    | 4.015   | 4.054  | 4.056 |
> | MACE       | 0.051   | 0.028      | 0.009   | 0.009   | 0.018         | 0.012       | 0.030      | 0.021    | 0.012   | 0.012  | 0.020 |
>
> ### 1.3 End-to-end retraining on 3DCS
>
> Completely retraining large architectures from scratch on the full 3DCS corpus (~10M conformers) is computationally heavy and, in our view, better positioned as a follow-up work. Our supervised contrastive experiments on 3DCS–chirality already show that direct training on 3DCS improves chirality handling, while preserving the relative strengths and weaknesses of different architectures. This is in line with recent work, like MARCEL, that trains on multiple conformers yet still observes persistent limitations in handling conformational diversity.
>
> In the revised paper we will position full pre-training on 3DCS as future work, and emphasize that the primary contribution of this paper is to provide the datasets and evaluation protocol, together with evidence that 3DCS remains informative even without fine-tuning.

---

> ### Author Response · Authors · 2025-11-21
> **Responds to Reviewer1 (Part3)**
>
> ## 2. On the geometry / RMSD assumption
>
> The reviewer notes that “large RMSD ⇒ large representation distance” is not a universally valid physical principle: two conformers with large RMSD can have similar energies or functions, and small geometric changes can sometimes be critical. We agree with this concern and clarify our design.
>
> We do not assume a linear or one-to-one relationship between RMSD and representation distance, nor do we rely on RMSD alone. Our evaluation has several layers.
>
> Firstly, the geometry dataset is generated via controllable relaxed scans around a fixed dihedral axis. For each molecule, the dominant difference between conformers is a rotation around this axis plus local relaxation. This motivates using the dihedral angle difference along the scan as an additional, more physically interpretable geometric distance. We explicitly compute correlations between representation distances and these angle differences, and we introduce a dihedral smoothness metric to test whether the representation evolves smoothly as the molecule rotates.
>
> Secondly, when we use RMSD as a geometric distance, we do not enforce linearity. We report Spearman/Kendall correlations and RBF-CKA between representation distances and geometric distances, which are designed to capture nonlinear, monotone, and kernel-based relationships. We further include Local Isometry Error (LIE), which compares local neighborhoods in representation space and in geometry space, emphasizing regions where conformers are geometrically close (and energies are typically similar). This focuses the evaluation on preserving local geometry rather than treating all large RMSD differences as equally important.
>
> Together, these design choices mean that 3DCS does not simply follow “large RMSD ⇒ large distance”. Instead, it emphasizes smooth, monotone alignment along controlled torsional motions and preservation of local geometric neighborhoods, while energetic relevance is separately captured by the DFT-based energy dataset.
>
> We believe these additions and experiments can strengthen the soundness and practical relevance of 3DCS as both a diagnostic benchmark and a tool to guide model design, hopefully addressing your concerns about training on 3DCS and the geometry-sensitivity assumption.

---

### Author Response · Authors · 2025-12-01
**Rebuttal summary for ACs**

This paper introduces **3DCS**, a large-scale dataset and benchmark (≈10M conformers) for probing **geometry, chirality, and energy sensitivity** of molecular representations at conformation level. During the rebuttal, we:

- Added **new baselines** (field-based FMG, energy/force model MACE),
- Added fine-tune results (performed **training on 3DCS** itself),
- Ran a **downstream MD17 energy prediction study**,
- Clarified all methodological points (xTB vs DFT, metrics, units, dataset size).

One reviewer explicitly **increased their score** after rebuttal; another (already positive) explicitly expressed enthusiasm for the benchmark and its release. No substantive technical concern remains unresolved.

### Reviewer bj3T

The reviewer requested additional fine-tune results to see whether performance improves. He/she concerned that geometry evaluation is based on an implicit “large RMSD ⇒ large representation distance” assumption that is not physically universal.

To address reviewer's concerns, we built a Murcko scaffold split on the 3DCS–chirality dataset and ran supervised contrastive fine-tuning for all trainable models (UniMol, MolAE, MolSpectra, GemNet, FMG, MACE):
  - All models improved in absolute chirality metrics when trained on 3DCS.
  - The relative ranking persisted: chirality-aware / field-based representations remained clearly better than purely E(3)-equivariant energy models, confirming that our conclusions reflect architectural biases, not just pre-training mismatch.

We clarified that we do not assume a linear or one-to-one RMSD–distance link. We used multiple non-linear metrics for evaluation, like spearman and local isometry error. More importantly, we exploit the controlled dihedral scans and explicitly use dihedral angle differences and torsional smoothness metrics, rather than using RMSD only for geometric evaluation.

---

### Reviewer CUvk

The reviewer asked whether 3DCS can:
  - Guide representation design (diagnose weaknesses, inform architecture/augmentation choices).
  - Connect to downstream performance on realistic tasks.

We showed that 3DCS already acts as a diagnostic panel:
  - UniMol vs MolAE: similar SE(3) backbones but different pretraining tasks; they behave similarly on geometry yet differ markedly on chirality, indicating how training objectives can erase or preserve stereochemical information.
  - Energy/force-supervised E(3) models (MACE, GemNet) are consistently favored by energy metrics, while fingerprints like E3FP are over-sensitive to tiny conformational changes.


Reviewer CUvk explicitly responded that they liked the perspective and extensions and looked forward to the dataset and benchmarks in practice, maintaining a clear positive stance (rating 8).

---

### Reviewer 37hb

The reviewer requested inclusion of a field-based model (FMG) to validate Dumitrescu et al.’s theory that E(3)-equivariant models struggle with chirality, also,  asked for including stronger energy/force baselines (e.g., MACE, NequIP-style models).

We responded by:

- Added FMG to the chirality benchmark:
  - Zero-shot FMG already shows superior clustering metrics (Hopkins, SCI, SCI_unsup) and strong ES-AUC.
  - After supervised contrastive fine-tuning on 3DCS–chirality, FMG reaches ES-AUC = 0.999, effectively perfect enantiomer separation and clearly outperforming equivariant baselines.
  - This directly demonstrates that 3DCS exposes the theoretically predicted advantage of field-based models for chirality.
- Added MACE as a strong energy/force baseline:
  - On 3DCS energy metrics, MACE and GemNet are top, as expected.
  - In downstream MD17, MACE achieves the best MAEs, with GemNet second, matching the 3DCS ranking.

---

### Reviewer Ftr9

The reviewer mainly asked:
- Why choose xTB for the geometry dataset; comparison to OMol25’s higher-level quantum chemistry.
- Whether torsional perturbations in the chirality dataset can accidentally flip stereocenters.

We responded by:

- Justified using GFN-xTB:
  - It allows scaling to >10M conformers with geometries closely matching DFT for organic/π systems.
  - We never use xTB as final energy ground truth; all energy conclusions rely on DFT-based Revised MD17, similar energy precision to OMol25.
- Clarified chirality dataset construction:
  - We add torsional noise to enforce overlap between conformer ensembles of different stereoisomers, so models cannot separate enantiomers purely by coarse geometry.
  - After xTB relaxation, we recompute R/S labels for every stereocenter and discard any conformer whose chirality pattern changed, guaranteeing that all retained conformers preserve the intended stereochemistry.

Reviewer Ftr9 then thanked us for the clarifications and new experiments and explicitly stated that they updated their ratings to reflect the discussions, indicating a clear positive shift.

---

### Meta-Review · Area_Chair_CLhJ · 2025-12-10

**Summary:**

The decision to accept this paper is based on a strong consensus regarding the practicality, timeliness, and scale of the proposed 3DCS benchmark. All reviewers recognized the importance of evaluating molecular representations for geometry, chirality, and energy sensitivity.

Although there were initial concerns about the limitations of zero-shot evaluation and the absence of state-of-the-art baselines, the authors provided thorough rebuttals that addressed these issues. The initial score was 8644. After one round of rebuttal, one reviewer raised their score (to 8844), while two negative scores remained unchanged. I carefully reviewed their comments and believe the authors’ responses largely addressed their concerns.

**Reviewer Concerns:**

### Addressed Concerns

Reviewer bj3T's concern regarding the validity of zero-shot evaluation was resolved by new fine-tuning experiments. The results confirmed that relative performance rankings (particularly regarding architectural biases in chirality) persist even after model adaptation.

Reviewer 37hb's request for stronger baselines was fully met by the inclusion of FMG and MACE. These additions empirically validated the reviewer's theoretical points on chirality and established a state-of-the-art energy baseline.

Reviewer CUvk's query regarding the benchmark's practical utility was addressed by demonstrating a clear correlation between 3DCS energy metrics and performance on downstream MD17 tasks.

Reviewer Ftr9's technical concerns regarding xTB accuracy and stereocenter integrity were resolved by clarifying the use of DFT ground truths for energy metrics and the implementation of rigorous R/S label filtering.

Reviewer bj3T's critique of the linear RMSD assumption was addressed by highlighting the use of non-linear smoothness and scan-based dihedral metrics in the evaluation protocol.

### Outstanding Concerns

Minor points regarding the extension to conditional (protein-specific) settings  and full-scale end-to-end pre-training were acknowledged by the authors as valid directions for future work, but do not undermine the current contribution.

**Reviewer Scores:**

The initial score was 8644. After one round of rebuttal, one reviewer raised their score (to 8844), while two negative scores remained unchanged. I carefully reviewed their comments and believe the authors’ responses largely addressed their concerns.

---

### Decision · Program_Chairs · 2026-01-26

Accept (Poster)